# EntroKV: Entropy-Guided Dynamic Budget Allocation for KV-Cache Compression

Wenhao Gao [1]  Haoran Cao [1]  Yueyan Li [1]  Yonggao Xiao [1]  Caixia Yuan [1]  Xiaojie Wang [1]

## Abstract

The prohibitive memory footprint of the Key-Value (KV) cache imposes a critical bottleneck for efficient long-context LLM serving. Current compression techniques typically rely on static or uniform budget allocation, overlooking the significant heterogeneity in information density across attention heads. To address this, we introduce ENTROKV, an entropy-driven dynamic budget allocation framework. Our method enables dynamic and rational allocation across layers, attention heads, and different tasks. We demonstrate that attention entropy serves as a robust proxy for compression sensitivity: heads with high entropy require larger retention budgets, whereas low-entropy heads can be aggressively compressed without accuracy degradation. Functioning as a lightweight, plug-and-play module, ENTROKV optimizes budget scheduling in real-time and is compatible with diverse compression operators. Extensive experiments demonstrate that ENTROKV consistently outperforms baselines, retaining ∼98% of full-cache performance at a 30% budget ratio with negligible computational overhead. Our code is available at https://github.com/Taylor-Gavel/EntroKV.

## 1. Introduction

Large Language Models (LLMs) rely heavily on the Key-Value (KV) cache mechanism when handling long-context tasks (Liu et al., 2025a; Wang et al., 2024a). KV cache mechanism is fundamental to the inference phase, enabling the model to retain and access extensive contextual information. However, as the context length increases, the linear expansion of the KV cache imposes severe memory and

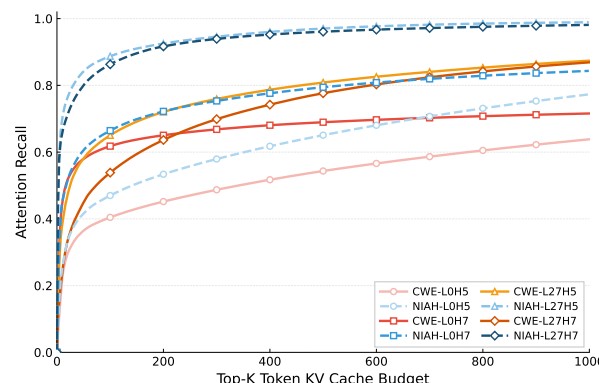

*Figure 1.* Substantial Heterogeneity in Attention Recall across Layers, Heads, and Tasks.

latency bottlenecks (Liu et al., 2023), as well as a degradation in generation quality (Liu et al., 2024b). Consequently, these limitations significantly hinder the practical deployment capabilities of current models.

To address these limitations, various KV cache compression strategies have been developed (LI et al., 2025). These primarily include *eviction*, which prunes tokens based on importance metrics (Li et al., 2024); *merging*, which aggregates redundant representations (Wan et al., 2025); and *quantization*, which reduces numerical precision (Liu et al., 2024c). However, these methods typically operate under the implicit assumption of uniform caching requirements across all model layers and attention heads. Such an assumption neglects the inherent structural heterogeneity of Transformers and the varying complexity ohf input contexts. Although *dynamic allocation* has recently been explored (Cai et al., 2024; Feng et al., 2024), a unified scheduling signal that remains stable across tasks, comparable across architectural components, and computable in online settings is still lacking.

To bridge this gap, we characterize the KV cache capacity requirements of individual components through the marginal utility of cache allocation. Specifically, we profile the *Attention Recall* of representative layer-head pairs as a function of the equivalent Top-$K$ budget. As illustrated in Figure 1, these utility curves exhibit substantial heterogeneity across both architectural and task dimensions. In the *Needle In A*

[1]Beijing University of Posts and Telecommunications. Correspondence to: Wenhao Gao <whgao@bupt.edu.cn>, Xiaojie Wang <xjwang@bupt.edu.cn>.

*Proceedings of the 43rd International Conference on Machine Learning*, Seoul, South Korea. PMLR 306, 2026. Copyright 2026 by the author(s).

*Haystack* (NIAH) task, the recall of most layer-head pairs saturates rapidly even under minimal budgets. Conversely, in the *Common Words Extraction* (CWE) task (Hsieh et al., 2024), saturation occurs more gradually, with different components exhibiting diverse growth rates and plateauing at significantly higher budget levels. These observations suggest that effective capacity requirements are jointly determined by structural position and task-specific semantics. Consequently, a uniform allocation strategy leads to a mismatch between resource supply and demand, causing redundancy in low-sensitivity components and information loss in high-sensitivity ones. This motivates an input-dependent allocation mechanism, driven by online signals, to maximize context fidelity under fixed resource constraints.

Building upon these observations, we propose EntroKV, an entropy-driven dynamic KV cache budget allocator. EntroKV leverages attention entropy as an online-computable metric of distribution dispersion to characterize the sensitivity of distinct attention heads to compression, thereby dynamically scheduling the retention budget for each head. This design effectively decouples the policy of "budget allocation" from the mechanism of "compression execution." Consequently, EntroKV functions as a plug-and-play front-end module that is naturally compatible with existing eviction- and merging-based operators, enabling fine-grained, content-aware budget scheduling without the need for additional training. Furthermore, EntroKV is mechanistically orthogonal to quantization: while EntroKV operates at the budget level (governing retention/aggregation intensity), quantization operates at the representation level (concerning precision and storage bit-width). These approaches can thus be composed to simultaneously achieve the dual benefits of dynamic scheduling and reduced memory footprint.

We summarize our main contributions as follows:

- We propose a novel perspective on long-context modeling that reframes KV cache compression as an information-aware budget allocation problem. This approach emphasizes the adaptive scheduling of resources across diverse components under strict constraints, moving beyond reliance on static heuristics.

- We introduce EntroKV, an entropy-guided, plug-and-play budget allocation framework. EntroKV leverages real-time attention entropy to dynamically determine the retention set size for individual attention heads and is designed to be compatible with both eviction-based and merging-based KV compression operators.

- We conduct extensive experiments on standard benchmarks, including LongBench and Ruler (Bai et al., 2023; Hsieh et al., 2024). The results demonstrate that EntroKV consistently enhances the generation quality of diverse compression operators while incurring

negligible computational overhead. This validates the effectiveness and versatility of our method as a universal, plug-and-play budget allocation module.

## 2. Related Work

### 2.1. KV Cache Compression

**Eviction and Merging.** KV cache compression methods reduce memory footprint by selectively discarding or aggregating tokens. Eviction-based approaches, including StreamLLM, H2O, Scissorhands, and SnapKV (Xiao et al., 2024; Zhang et al., 2023; Liu et al., 2023; Li et al., 2024), retain tokens based on attention-derived importance, positional priors, or snapshot heuristics. Complementarily, merging-based methods such as CaM, KVMerger, and D2O (Zhang et al., 2024; Wang et al., 2024b; Wan et al., 2025) consolidate semantically similar KV pairs to mitigate information loss, with recent extensions exploring depth-aware or discriminative merging strategies (Liu et al., 2024a; 2025b). Overall, these methods focus on *how* to compress KV caches under a given budget, while typically assuming uniform or fixed allocation strategies across layers and heads.

**Quantization.** Quantization-based approaches (e.g., KIVI, KVQuant, and Atom (Liu et al., 2024c; Hooper et al., 2024; Zhao et al., 2024)) reduce KV cache memory usage by lowering numerical precision. These methods are largely orthogonal to token-level compression and budget allocation strategies, and can be combined with eviction or merging techniques. In this work, we focus on allocating token retention budgets and do not consider quantization-based approaches.

### 2.2. Dynamic Budget Allocation

Recent studies have explored dynamic KV cache budget allocation beyond static heuristics. Layer-wise strategies, such as PyramidKV, SqueezeAttention, and DynamicKV (Cai et al., 2024; Wang et al., 2025; Zhou et al., 2025), adjust budgets based on coarse-grained sensitivity patterns across depth. AdaKV (Feng et al., 2024) further extends this idea to the attention-head level, while LAVa (Shen et al., 2025) investigates joint layer–head optimization via output-loss surrogates. Task-KV (He et al., 2025) leverages semantic differentiation to allocate budgets at the task level, and CAKE (Qin et al., 2025) combines layer-level entropy with temporal variance for cascading redistribution. Despite their effectiveness, existing methods typically rely on handcrafted heuristics, auxiliary computation branches, or task-specific signals, and lack a unified, comparable, and online-computable metric for fine-grained budget scheduling across diverse inputs and architectural components.

## 3. EntroKV

We propose EntroKV, a dynamic KV cache budget allocation framework guided by attention entropy. We first formulate KV cache compression as a constrained resource allocation problem (§3.1), then analyze properties of effective allocation signals and motivate entropy as an actionable proxy (§3.2). We subsequently present the algorithm (§3.3) and show how EntroKV integrates with diverse compression operators (§3.4).

### 3.1. Problem Formulation: KV Cache Compression as Resource Allocation

Research on KV cache compression generally falls into two orthogonal directions: compression operator design (addressing how to compress) and budget allocation strategies (determining how much to compress). The former aims to minimize information loss under a fixed budget constraint, whereas the latter focuses on the optimal allocation of limited resources across diverse components. This work focuses on the latter.

Consider a Transformer model with $L$ layers and $H$ attention heads per layer. During inference, given a total retention budget $B$ (representing the system's total capacity for KV pairs), our objective is to determine an allocation scheme $\{B_l^h\}_{l\in[L],h\in[H]}$ that minimizes the overall approximation error induced by compression, subject to the budget constraint:

$$\min_{\{B_l^h\}} \sum_{l=1}^{L} \sum_{h=1}^{H} \mathcal{L}_l^h(B_l^h) \quad \text{s.t.} \quad \sum_{l=1}^{L} \sum_{h=1}^{H} B_l^h \leq B \quad (1)$$

where $\mathcal{L}_l^h(B_l^h)$ denotes the *compression loss* of the $h$-th head in the $l$-th layer under budget $B_l^h$.

**Assumption (Diminishing Returns).** In practice, allocating more KV capacity to a head does not worsen performance, and the marginal benefit typically decreases as the budget grows. Thus we assume $\mathcal{L}_l^h(B)$ is non-increasing in $B$ with diminishing marginal gains. This assumption holds across common eviction/merging operators and supports the need for differentiated allocation.

Prevailing methods typically employ **a uniform allocation strategy**, assigning an identical retention budget $B_l^h = B/(L \cdot H)$ to every attention head. However, as illustrated in Figure 1, the marginal utility of cache capacity exhibits significant heterogeneity across different layer-head pairs: some saturate rapidly even with a minimal budget, whereas others continue to yield substantial marginal gains over a wider budget range. Given such heterogeneity, a uniform allocation strategy is suboptimal for minimizing the global loss—it inevitably squanders resources on "easy-to-compress" components while incurring excessive information loss in "hard-to-compress" ones. This motivates the

following question: **How can we efficiently achieve differentiated allocation during inference under a strict global budget?**

### 3.2. Motivation and Proxy Signal Analysis

Directly optimizing Eq. (1) during inference is challenging: (i) $\mathcal{L}_l^h(\cdot)$ is generally non-differentiable w.r.t. allocation for discrete compression operators, (ii) it is only observable post-hoc after generating outputs, and (iii) it is difficult to decompose into actionable per-head signals. We therefore seek an online-computable proxy that predicts component-wise compression sensitivity prior to executing compression. Our goal is to identify a signal that is stable, comparable across heads, and empirically predictive of recall degradation under compression.

**Attention Recall as a Proxy Objective.** KV cache compression approximates attention computation. For Top-$K$ retention-style operators, we define total attention recall as:

$$\text{Recall}_{\text{total}} = \frac{1}{L \cdot H} \sum_{l=1}^{L} \sum_{h=1}^{H} \sum_{i \in \text{Top-}K_l^h} a_i^{(l,h)}, \quad (2)$$

where $a_i^{(l,h)}$ denotes the attention weight assigned to position $i$ by head $h$ in layer $l$. Higher recall means more attention mass is preserved after compression and thus typically implies lower approximation error. Figure 2(a) shows that recall correlates with downstream loss across diverse configurations. **Notably, at similar recall levels, tighter budgets can achieve lower loss than looser budgets.** This indicates that simply increasing a *uniform* budget can preserve redundant mass without proportionally reducing loss, reinforcing the need for selective, differentiated allocation.

However, recall is an a posteriori quantity computed after compression and thus cannot directly guide allocation beforehand.

**Attention Entropy as an Actionable Signal.** We next connect compression sensitivity to the *shape* of attention distributions. Intuitively, "peaky" distributions are easier to compress since Top-$K$ captures most probability mass, while "flat" distributions are harder because a large long-tail mass is discarded. Shannon entropy $H(\mathbf{a}) = -\sum_i a_i \log a_i$ quantifies dispersion: low entropy corresponds to concentrated attention and high entropy corresponds to diffuse attention.

**Oracle construction for Figure 2(b).** To contextualize the "near-oracle" behavior, we construct an *offline upper bound* by assuming access to a global importance score for every candidate cache entry. Concretely, we compute an importance score $s_{l,h,i}$ for each entry (e.g., the $i$-th cached position associated with head $h$ in layer $l$), concatenate all scores across layers and heads into a single list, and

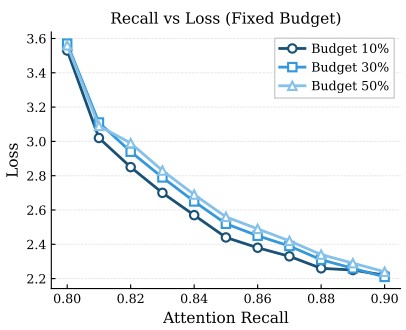
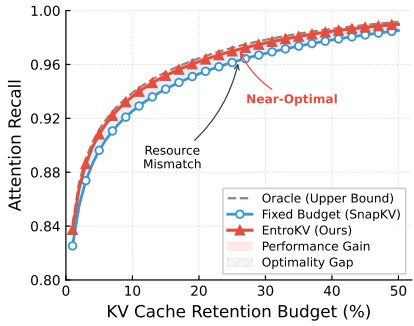
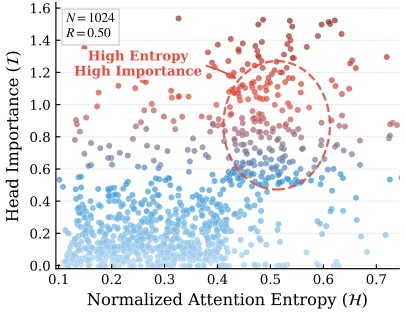

*(a)* Loss vs. Recall under fixed budgets from 10% to 50%.

*(b)* EntroKV approaches an oracle upper bound.

*(c)* High-entropy heads correlate with high importance.

*Figure 2.* **Entropy as an Effective Signal for KV Cache Budget Allocation. (a)** At similar attention recall levels, tighter budgets (e.g., 10%) can yield lower loss than looser budgets (e.g., 50%), suggesting that unreasonable allocation at high budgets wastes capacity on redundant components. **(b)** EntroKV approaches an *offline oracle* upper bound constructed by globally selecting the top-$B$ most important KV entries across all layers and heads (details in §3.2).. **(c)** Normalized attention entropy correlates with Taylor head importance.

then *globally* retain the top entries with the highest scores under the same total budget constraint (i.e., selecting the top-$B$ entries). This oracle corresponds to a hindsight global selector that is not available online, but provides a strong upper bound reference. As shown in Figure 2(b), EntroKV substantially improves over uniform allocation and closely approaches this offline upper bound across budgets. This oracle corresponds to a hindsight global selector that is not available online and does not constitute an achievable optimum, but serves as a diagnostic upper reference.

**Semantic Importance of High-Entropy Heads.** We further observe that entropy is aligned with a notion of head importance. Figure 2(c) plots normalized entropy against Taylor importance (Molchanov et al., 2019):

$$I_l^h = \left| \frac{\partial \mathcal{L}}{\partial \mathbf{O}_l^h} \odot \mathbf{O}_l^h \right|_F, \tag{3}$$

where $\mathbf{O}_l^h$ is the head output and $\mathcal{L}$ is the task loss. The positive correlation suggests that high-entropy heads often encode more critical contextual dependencies, so allocating them more capacity can be beneficial not only for recall but also for preserving semantically important computation paths.

**Summary.** The analysis above establishes a coherent logic chain: attention entropy serves as a proxy for both compression difficulty and semantic importance. High-entropy heads exhibit diffuse attention distributions, requiring larger budgets to preserve comparable recall; meanwhile, they tend to encode more critical contextual dependencies. Consequently, entropy-guided allocation naturally prioritizes budget towards heads that are both hard to compress and semantically important, enabling near-optimal resource scheduling without requiring offline profiling or task-specific tuning.

### 3.3. Algorithm Design

---

**Algorithm 1** EntroKV (budget allocation front-end)

**Require:** Total budget ratio $\beta$, adaptivity $\alpha$, baseline $\bar{H}$, backend operator $\mathcal{C}$
1: **for** each compression step (e.g., each observation window) with length $m$ **do**
2:     Obtain attention weights $\{\mathbf{a}_l^h\}$ from backend (no extra matmul)
3:     Compute $\tilde{H}_l^h = H(\mathbf{a}_l^h)/\log m$ for all heads
4:     Set $B_l^h \leftarrow \beta[(1-\alpha)\bar{H} + \alpha\tilde{H}_l^h]$
5:     Convert to operator-specific control (e.g., $K_l^h = \lfloor B_l^h \cdot m \rfloor$)
6:     Apply backend compression $\mathcal{C}$ using $\{K_l^h\}$ (or target ratios)
7: **end for**

---

Building upon the above motivation, EntroKV dynamically computes entropy to govern head-wise budget allocation during inference.

**Normalized Entropy.** Since raw entropy scales with sequence length $n$ ($H_{\max} = \log n$), we use normalized attention entropy:

$$\tilde{H}_l^h = \frac{H(\mathbf{a}_l^h)}{\log n} \in [0, 1]. \tag{4}$$

In practice, the entropy is computed over the last $w$ queries in the backend's observation window (e.g., SnapKV), reusing attention weights already materialized during the forward pass without introducing additional matrix multiplications. A small window ($w \geq 16$) suffices to reliably estimate compression sensitivity (see Appendix E.1 for the full formulation and Appendix I for window-length ablations).

**Global Baseline (Calibration).** We estimate a global reference entropy on a representative calibration set (details in Appendix D):

$$\bar{H} = \frac{1}{N} \sum_{t=1}^{N} \frac{1}{L \cdot H} \sum_{l=1}^{L} \sum_{h=1}^{H} \tilde{H}_l^h(t). \tag{5}$$

*Table 1.* Long-Context Benchmark Accuracy under Fixed and Entropy-Guided KV Cache Budget Allocation (Retention Ratio $\leq 0.3$).

| | LLAMA-3.1-8B-INSTRUCT | | | QWEN2.5-7B-INSTRUCT | | | QWEN3-8B | | |
| --- | --- | --- | --- | --- | --- | --- | --- | --- | --- |
| | LONGBENCH | RULER-4K | RULER-16K | LONGBENCH | RULER-4K | RULER-16K | LONGBENCH | RULER-4K | RULER-16K |
| FULL CACHE | $48.26 \pm 0.15$ | $95.64 \pm 0.07$ | $91.73 \pm 0.20$ | $48.47 \pm 0.36$ | $94.61 \pm 0.05$ | $90.83 \pm 0.15$ | $49.68 \pm 0.17$ | $95.33 \pm 0.12$ | $93.12 \pm 0.13$ |
| SNAPKV | $47.01 \pm 0.18$ | $86.91 \pm 0.05$ | $90.88 \pm 0.08$ | $47.34 \pm 0.24$ | $93.01 \pm 0.11$ | $89.21 \pm 0.16$ | $48.25 \pm 0.10$ | $93.47 \pm 0.08$ | $91.21 \pm 0.08$ |
| **ENTRO-SNAPKV** | **$48.15 \pm 0.15$** | **$89.33 \pm 0.27$** | **$91.63 \pm 0.30$** | **$48.13 \pm 0.18$** | **$93.38 \pm 0.10$** | **$89.37 \pm 0.22$** | **$49.05 \pm 0.08$** | **$95.26 \pm 0.08$** | **$92.89 \pm 0.11$** |
| CAM | $47.31 \pm 0.11$ | $83.70 \pm 0.15$ | $67.18 \pm 0.15$ | $46.93 \pm 0.16$ | $91.97 \pm 0.14$ | $84.67 \pm 0.12$ | $48.84 \pm 0.15$ | $93.58 \pm 0.10$ | $91.24 \pm 0.07$ |
| **ENTRO-CAM** | **$49.02 \pm 0.13$** | **$89.21 \pm 0.17$** | **$91.57 \pm 0.11$** | **$48.77 \pm 0.11$** | **$94.34 \pm 0.14$** | **$90.53 \pm 0.19$** | **$49.55 \pm 0.14$** | **$95.35 \pm 0.09$** | **$92.99 \pm 0.12$** |
| D2O | $47.81 \pm 0.20$ | $85.88 \pm 0.16$ | $74.71 \pm 0.19$ | $41.23 \pm 0.24$ | $87.68 \pm 0.14$ | $82.67 \pm 0.21$ | $48.51 \pm 0.10$ | $93.31 \pm 0.03$ | $91.41 \pm 0.09$ |
| **ENTRO-D2O** | **$49.11 \pm 0.21$** | **$88.51 \pm 0.21$** | **$90.87 \pm 0.19$** | **$45.89 \pm 0.15$** | **$91.08 \pm 0.24$** | **$86.42 \pm 0.27$** | **$49.73 \pm 0.10$** | **$95.21 \pm 0.03$** | **$93.09 \pm 0.05$** |

**Implementation note.** In our implementation, $\bar{H}$ is computed once on a small held-out calibration workload and fixed thereafter. When the deployment domain shifts significantly, recomputing $\bar{H}$ can improve stability, especially for large $\alpha$.

**Dynamic Allocation Rule.** Given an input, we allocate budget to each head $(l, h)$ as:

$$B_l^h = \beta \cdot \left[ (1 - \alpha) \cdot \bar{H} + \alpha \cdot \tilde{H}_l^h \right], \qquad (6)$$

where $\alpha \in [0, 1]$ trades off global stability vs. instance adaptivity, and $\beta > 0$ controls the overall retention rate.

**Interpretation.** Let $\Delta = \alpha(\tilde{H}_l^h - \bar{H})$, then $B_l^h = \beta(\bar{H} + \Delta)$. Heads with $\tilde{H}_l^h > \bar{H}$ receive additional budget, while low-entropy heads release budget for redistribution.

**Algorithmic schedule (online inference).** EntroKV updates budgets at the same granularity as the backend operator. For eviction methods with an observation window (e.g., SnapKV), entropy and budgets are updated *once per window*, reusing the attention matrices already computed within that window.

**Complexity.** Entropy computation is a lightweight reduction over existing attention weights (e.g., $\sum_i a_i \log a_i$). When using windowed attention (as in SnapKV), the additional overhead is $O(L \cdot H \cdot m)$ per window of size $m$, which is negligible compared to the attention computation cost.

### 3.4. Framework Integration

EntroKV decouples **budget allocation** (how much to compress) from **compression execution** (how to compress), enabling plug-and-play integration with diverse operators.

**Integration with Eviction-based Operators.** Eviction methods (e.g., SnapKV, H$_2$O, TOVA) (Li et al., 2024; Zhang et al., 2023; Oren et al., 2024) discard tokens based on importance scores. EntroKV integrates via: (1) reusing attention weights computed within the observation window to extract entropy signals; (2) replacing static uniform $K$ with dynamic head-wise budgets $K_l^h = \lfloor B_l^h \cdot m \rfloor$ (with $m$ being the step/window length); and (3) preserving compat-

ibility with FlashAttention and chunk-wise inference. For SnapKV, we keep its observation window and pooling design unchanged and only modify the budget computation logic.

**Integration with Merging-based Operators.** Merging methods (e.g., CaM, D2O) (Zhang et al., 2024; Wan et al., 2025) aggregate similar KV pairs, controlled by target lengths or merge ratios. EntroKV interprets $B_l^h$ as a head-wise target retention ratio: high-entropy heads use weaker merging (higher retention), while low-entropy heads allow more aggressive merging.

## 4. Experiment

### 4.1. Settings

**Baseline.** We evaluate the effectiveness of our proposed allocation strategy across various KV cache compression paradigms, including eviction-based methods (e.g., SnapKV (Li et al., 2024)) and merging-based methods (e.g., CaM and D2O (Zhang et al., 2024; Wan et al., 2025)). Furthermore, we conduct a comparative analysis against contemporary dynamic allocation frameworks, including PyramidKV, AdaKV, LAVa, Task-KV, and CAKE (Cai et al., 2024; Feng et al., 2024; Shen et al., 2025; He et al., 2025; Qin et al., 2025).

**Foundation Model.** We select several representative open-source models for our evaluation, namely Llama-3.1-8B-Instruct, Qwen2.5-7B-Instruct, and Qwen3-8B (Grattafiori et al., 2024; Yang et al., 2024; 2025).

**Benchmarks.** To demonstrate the effectiveness of our proposed method, we conduct extensive evaluations on long-context benchmarks, including LongBench and RULER (Bai et al., 2023; Hsieh et al., 2024), as well as several short-text datasets (Hendrycks et al., 2021; Rajpurkar et al., 2018).

**Others.** All experiments are conducted on NVIDIA A800 (80GB) GPUs. We report the mean results averaged over five independent runs, alongside standard deviations to demonstrate the stability of our method's improvements.

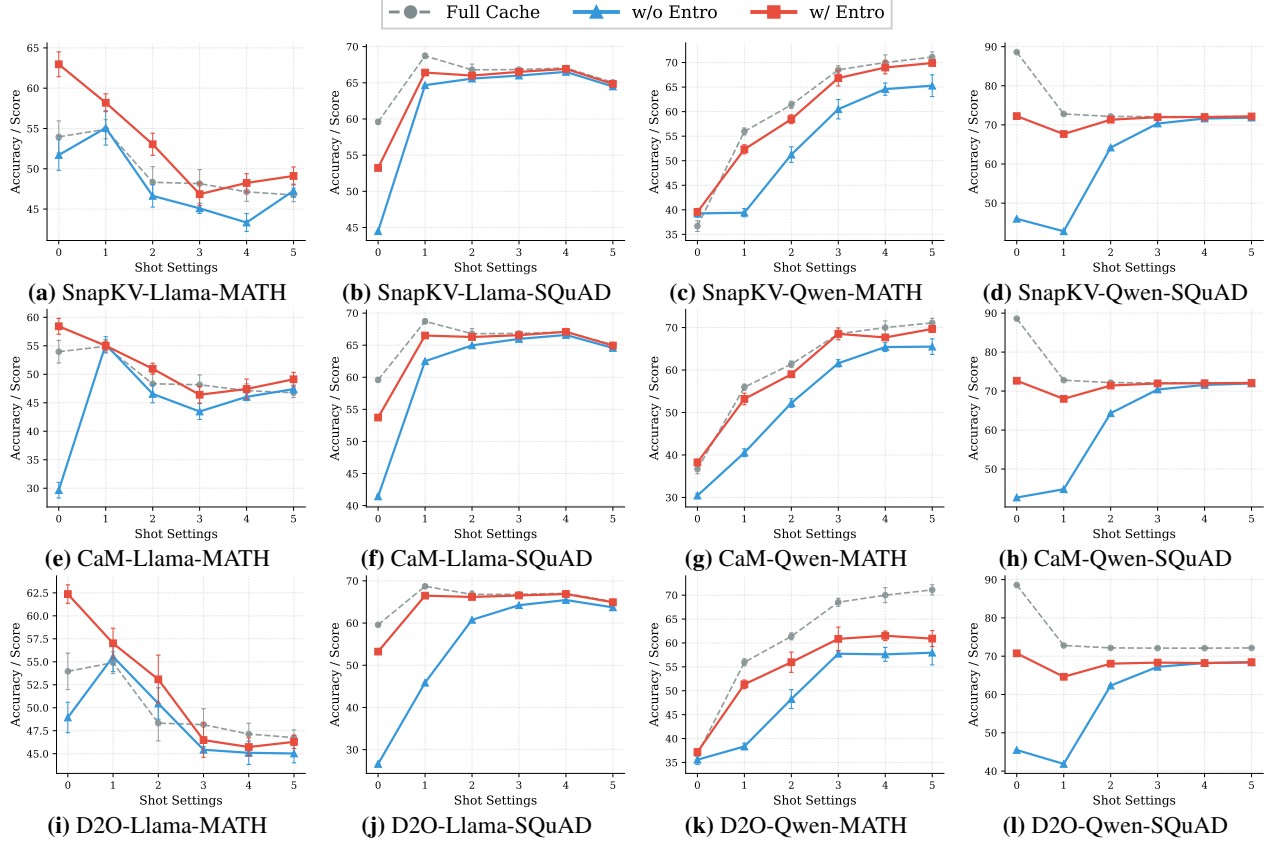

**Figure 3. Entropy as a Robust Signal for Adaptive KV Cache Budget Allocation.** Performance under different shot settings across two models (Llama, Qwen), three KV cache allocation methods (SnapKV, CaM, D2O), and two datasets (MATH, SQuAD).

Detailed hyperparameters and prompt templates are provided in Appendix C.2.

### 4.2. Accuracy

We report evaluation results across both long-context and short-context benchmarks to validate the effectiveness of our proposed method. We demonstrate that our approach consistently outperforms both fixed-budget allocation baselines and contemporary dynamic allocation frameworks.

**Long-Context Benchmarks.** As shown in Table 1, we evaluate EntroKV on LongBench and RULER benchmarks across three representative foundation models, including the recent Qwen3-8B. Several key observations emerge from the results. **First**, EntroKV consistently improves all baseline compression methods across all three models and benchmarks, demonstrating its effectiveness as a universal budget allocation module. **Second**, the integration of EntroKV enables compressed models to approach or even match the performance of Full Cache inference, particularly on the LongBench benchmark, validating our theoretical motivation that entropy-guided allocation better preserves critical information. **Third**, the performance gains are more pro-

nounced for merging-based operators (CaM and D2O) compared to eviction-based methods (SnapKV), suggesting that dynamic allocation is especially beneficial when the underlying compression operator introduces greater information distortion. **Fourth**, the improvements generalize across different context lengths (4K and 16K in RULER), indicating that our entropy-based signal remains effective regardless of sequence scale. **Fifth**, EntroKV seamlessly generalizes to recent model architectures (Qwen3-8B), consistently enhancing all underlying compression operators and approaching Full Cache performance without any architecture-specific tuning. **Finally**, the small standard deviations across all configurations confirm that EntroKV provides stable and consistent improvements rather than incidental gains.

**Compared with other Budget Allocation Methods.** Table 2 compares EntroKV with a comprehensive set of dynamic allocation methods, including PyramidKV, AdaKV, LAVa, Task-KV, and CAKE (Cai et al., 2024; Feng et al., 2024; Shen et al., 2025; He et al., 2025; Qin et al., 2025), under the same SnapKV compression operator. Entro-SnapKV consistently outperforms all baselines across LongBench and both RULER settings. While PyramidKV and AdaKV

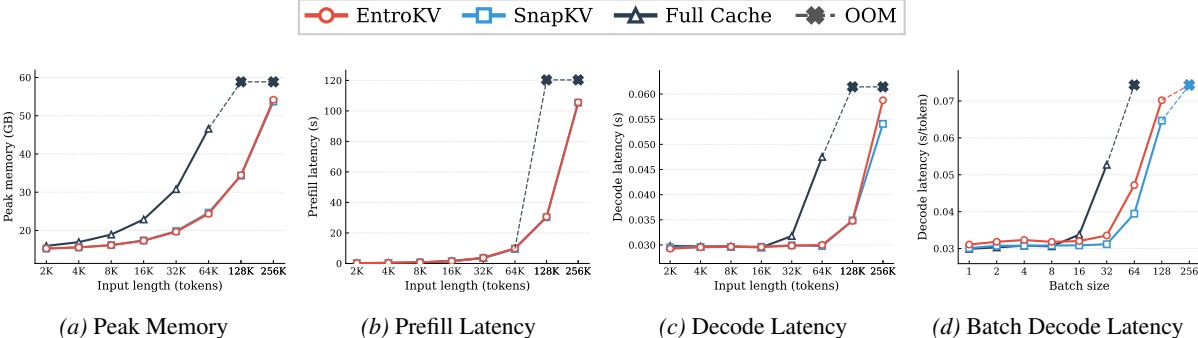

*(a)* Peak Memory *(b)* Prefill Latency *(c)* Decode Latency *(d)* Batch Decode Latency

*Figure 4.* **Efficiency Analysis of EntroKV.** We evaluate the memory usage and decoding latency under varying context lengths and batch sizes. (a) Peak GPU memory usage as the context length increases. (b) Prefill latency under increasing context length. (c) Decode latency under increasing context length. (d) Batch decode latency under increasing batch sizes (2K context length per batch).

provide only marginal improvements over static SnapKV, and recent methods such as LAVa and Task-KV even underperform on RULER-16K, EntroKV yields substantially larger gains across all benchmarks. Compared to CAKE—which also leverages entropy but at a coarser layer-level granularity—EntroKV achieves superior performance by performing fine-grained head-level allocation with a single lightweight reduction over existing attention weights, requiring no auxiliary computation branches. A detailed method-level comparison of proxy signals, allocation granularity, and computational overhead is provided in Appendix H.

**Short-Context Benchmarks.** While EntroKV is primarily designed for long-context KV bottlenecks, we further evaluate its effectiveness in *short-to-mid* context regimes. We conduct experiments on **MATH-500** (Hendrycks et al., 2021) and **SQuAD** (Rajpurkar et al., 2018), progressively increasing in-context demonstrations (0–5 shot) to enlarge the effective prompt length. As shown in Figure 3, across both **Llama** and **Qwen** backbones and three compression operators (SnapKV, CaM, D2O), EntroKV consistently improves the corresponding fixed-budget baselines under the same compression ratio, indicating that attention entropy serves as a robust signal for head-wise budget allocation beyond long-context settings.

Notably, **Full Cache** performance does not monotonically improve with additional demonstrations. In several MATH configurations (e.g., subplots (a), (e), and (i)), accuracy degrades as more examples are added, suggesting sensitivity to redundant or distracting prompt content. In contrast, entropy-guided allocation remains stable and can even surpass Full Cache performance under larger few-shot settings, implying that selective retention guided by entropy may implicitly suppress noisy or less informative attention patterns.

*Table 2.* Comparison of Different Budget Allocation Methods in Llama-3.1-8B-Instruct.

| | LONGBENCH | RULER-4K | RULER-16K |
|---|---|---|---|
| SNAPKV | $47.01 \pm 0.18$ | $86.91 \pm 0.05$ | $90.88 \pm 0.08$ |
| PYRAMIDKV | $47.30 \pm 0.25$ | $86.94 \pm 0.05$ | $90.63 \pm 0.19$ |
| ADA-SNAPKV | $47.79 \pm 0.18$ | $86.40 \pm 0.34$ | $90.33 \pm 0.34$ |
| ADA-PYRAMIDKV | $47.69 \pm 0.10$ | $86.85 \pm 0.12$ | $91.35 \pm 0.12$ |
| LAVA | $47.25 \pm 0.19$ | $85.24 \pm 0.24$ | $87.12 \pm 0.20$ |
| TASK-KV | $46.66 \pm 0.11$ | $86.45 \pm 0.18$ | $87.41 \pm 0.22$ |
| CAKE | $47.24 \pm 0.19$ | $88.02 \pm 0.13$ | $91.02 \pm 0.30$ |
| ENTRO-SNAPKV | $\mathbf{48.15 \pm 0.15}$ | $\mathbf{89.33 \pm 0.27}$ | $\mathbf{91.63 \pm 0.30}$ |

### 4.3. Efficiency

We evaluate the efficiency of EntroKV in terms of memory footprint and decoding latency.

**Latency and Memory Efficiency.** Figure 4(a) reports the peak GPU memory usage as the context length increases. The Full Cache baseline exhibits linear memory growth and triggers Out-Of-Memory (OOM) errors at large contexts, whereas EntroKV maintains a significantly lower and bounded memory footprint. Figures 4(b) and 4(c) show the prefill and decode latency, respectively. EntroKV closely follows the latency trends of SnapKV in both phases, indicating that entropy-based budget allocation introduces negligible computational overhead.

**Batch Decode Latency.** Figure 4(d) evaluates decoding latency under increasing batch sizes. The Full Cache baseline encounters OOM at batch size 64 due to excessive KV cache memory usage, while EntroKV significantly reduces memory consumption and enables batch decoding to scale up to batch size 128 with stable latency.

### 4.4. Ablation

This section presents an ablation study on the primary hyperparameters $\alpha$ and $\beta$, as well as the impact of context length scaling, to investigate their effects on allocation per-

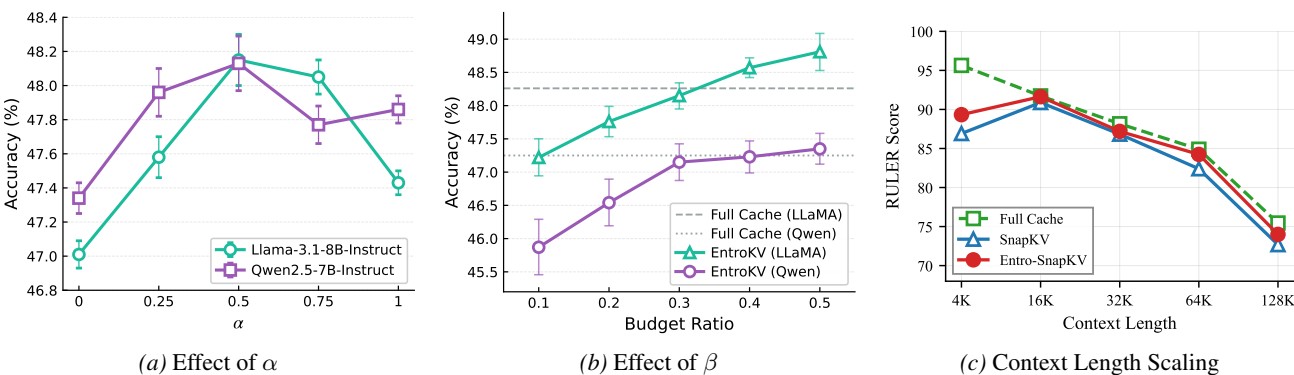

*(a)* Effect of $\alpha$  *(b)* Effect of $\beta$  *(c)* Context Length Scaling

*Figure 5.* **Ablation Study.** (a) Effect of adaptive coefficient $\alpha$. (b) Effect of total budget ratio $\beta$. (c) RULER performance across context lengths (4K–128K) on Llama-3.1-8B-Instruct.

formance. Furthermore, we provide a detailed discussion and qualitative observations gathered during our experimental evaluation to offer deeper insights into the method's behavior.

**Effect of Adaptive Coefficient $\alpha$.** Figure 5(a) illustrates the impact of the adaptive coefficient $\alpha$ on downstream task performance across multiple foundation models. Recall that $\alpha$ governs the trade-off between global statistics ($\bar{H}$) and instance-specific entropy ($\tilde{H}_l^h$) in budget allocation. We observe a consistent inverted-U pattern: performance initially improves as $\alpha$ increases from 0, reaches a peak around $\alpha = 0.5$, and subsequently degrades as $\alpha$ approaches 1. This behavior admits an intuitive explanation. When $\alpha = 0$, the allocation reduces to a static strategy that ignores input-specific variations, failing to adapt to the heterogeneous compression requirements of different contexts. Conversely, when $\alpha = 1$, the allocation relies entirely on instantaneous entropy estimates, which may introduce instability due to noise in individual attention patterns. The optimal value at $\alpha \approx 0.5$ suggests that effective budget allocation requires balancing the stability provided by global calibration statistics with the adaptivity afforded by real-time entropy signals. Notably, both Llama-3.1-8B-Instruct and Qwen2.5-7B-Instruct exhibit similar optimal $\alpha$ values, indicating that this hyperparameter generalizes well across different model architectures.

**Effect of Total Budget $\beta$.** Figure 5(b) examines the relationship between the overall budget ratio (controlled by $\beta$) and model accuracy. As expected, performance improves monotonically with increasing budget allocation, as more KV pairs are retained to preserve contextual information. Notably, EntroKV demonstrates strong performance even under aggressive compression settings: at a budget ratio of 0.2, Llama-3.1-8B-Instruct achieves approximately 98% of Full Cache performance, while at 0.3, it nearly matches the uncompressed baseline. More strikingly, at higher budget

ratios (0.4–0.5), EntroKV on Llama actually *exceeds* Full Cache accuracy, suggesting that selective retention guided by entropy may implicitly filter out noisy or redundant tokens that would otherwise dilute attention. For Qwen2.5-7B-Instruct, the gains are more gradual but consistent, converging toward Full Cache performance as the budget increases. These results validate that EntroKV effectively allocates limited resources to maximize information retention, enabling practitioners to select appropriate compression rates based on their memory-accuracy trade-off requirements.

**Effect of Context Length Scaling.** Figure 5(c) evaluates EntroKV's robustness across context lengths ranging from 4K to 128K on the RULER benchmark (Llama-3.1-8B-Instruct). As context length increases, all methods exhibit performance degradation, reflecting the inherent difficulty of long-context reasoning. However, Entro-SnapKV consistently outperforms the SnapKV baseline at every context length and closely tracks the Full Cache upper bound. Notably, even at 128K—far beyond the 16K context evaluated in Table 1—EntroKV preserves 98.1% of Full Cache performance (74.00 vs. 75.45), compared to 96.3% for SnapKV (72.68 vs. 75.45). This demonstrates that attention entropy remains a highly resilient scheduling signal under extreme context scaling, and the entropy-guided allocation strategy does not degrade or require recalibration as the input sequence grows substantially longer.

**Discussion: Implicit Denoising via Entropy-Guided Allocation.** The observation that EntroKV can exceed Full Cache performance may initially appear contradictory to the diminishing-returns assumption in Section 3.1, which states that allocating more budget to a single head does not worsen its local approximation quality. We clarify that this is not a mathematical contradiction but rather reflects a subtle distinction between *microscopic* approximation error and *macroscopic* downstream task performance. The monotonic non-increasing assumption for local compression loss

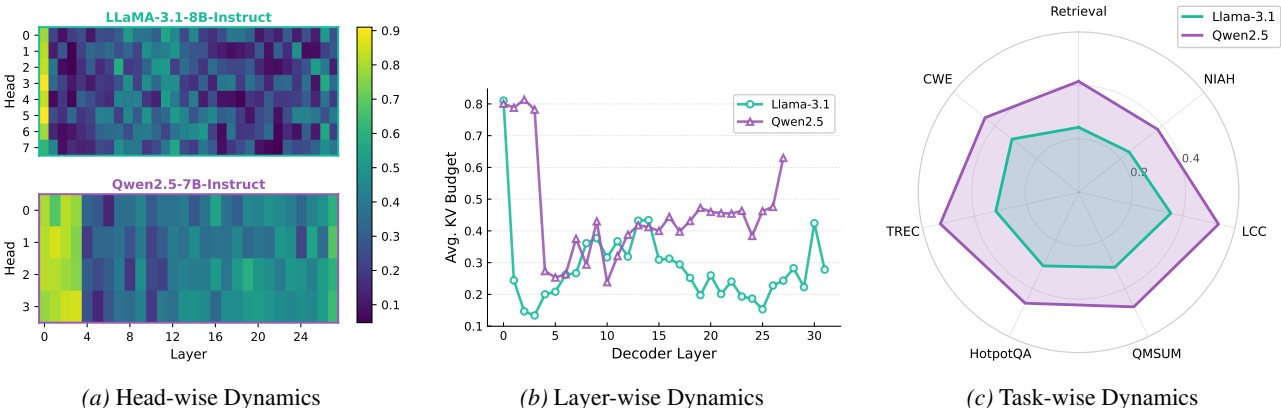

*(a)* Head-wise Dynamics    *(b)* Layer-wise Dynamics    *(c)* Task-wise Dynamics

*Figure 6.* **EntroKV Budget Allocation Varies Across Heads, Layers, and Tasks.** (a) Heatmap of budget allocation across all layer–head pairs; brighter colors indicate larger budgets. (b) Layer-averaged budget follows architecture-specific non-monotonic patterns. (c) Task-wise allocation adapts to workload characteristics, with retrieval-heavy tasks receiving larger budgets.

correctly dictates that a larger budget helps an individual head better approximate its Full Cache behavior. However, perfectly reconstructing Full Cache outputs for all heads does not guarantee optimal downstream accuracy, because LLMs are highly susceptible to noise in long contexts (Liu et al., 2024b)—redundant or distracting tokens captured by the Full Cache can actively mislead the model.

EntroKV's allocation mechanism provides an implicit denoising effect: by assigning minimal budgets to low-entropy heads—which typically attend to noisy or redundant content—the framework prevents reconstructing potentially harmful Full Cache attention patterns in those components, while concentrating resources on high-entropy heads that encode critical contextual dependencies. This selective compression effectively functions as an information bottleneck, filtering noise while preserving key evidence. As empirical support, Figure 2(a) shows that at similar attention recall levels, tighter budgets (10%) can yield lower downstream loss than looser budgets (50%), indicating that forcing complete attention recovery may inadvertently introduce harmful noise rather than improve generation quality.

### 4.5. Observation

To gain further insight into how entropy-guided allocation operates in practice, Figure 6 visualizes the budget distribution produced by EntroKV across heads, layers, and tasks. The results reveal that EntroKV adaptively responds to the underlying heterogeneity in compression sensitivity.

**Layer–Head Allocation Heatmap.** Figure 6(a) presents a heatmap of EntroKV's budget allocation across all layer–head pairs. The results exhibit pronounced heterogeneity: certain layer–head combinations consistently receive high budgets (bright regions), while others are assigned minimal budgets (dark regions). This fine-grained allocation

confirms that uniform budget strategies fail to capture the diverse compression requirements across the model.

**Layer-wise Allocation.** As shown in Figure 6(b), the average budget allocated by EntroKV follows a non-monotonic pattern across layers. This observation challenges fixed layer-based heuristics (e.g., monotonically decreasing budgets with depth) and demonstrates that EntroKV adapts to architecture-specific sensitivity profiles without manual tuning.

**Task-wise Allocation.** Figure 6(c) illustrates that EntroKV's allocation adapts to task semantics. Retrieval-intensive tasks induce higher average budgets than tasks with greater input redundancy, indicating that entropy effectively captures task-dependent compression difficulty.

## 5. Conclusion

We present ENTROKV, a plug-and-play framework using attention entropy to optimize resource allocation for KV cache compression methods in long-context LLMs. We show that entropy reliably reflects compression sensitivity, enabling accurate identification of heavy-hitter heads. By decoupling budget allocation from compression execution, ENTROKV functions as a plug-and-play module that consistently improves diverse compression strategies (e.g., eviction and merging) with negligible overhead. Our results highlight the importance of content-aware resource scheduling for balancing efficiency and accuracy in large-scale LLM deployment. Future work includes extending entropy-guided allocation to multimodal LLMs and co-designing with quantization for compounded memory savings.

## Acknowledgements

We would like to thank anonymous reviewers for their suggestions and comments sincerely. The work was supported by the Beijing Natural Science Foundation (L247010).

## Impact Statement

This paper presents work whose goal is to advance the field of foundational LLM inference efficiency. There are many potential societal consequences of our work, none of which we feel must be specifically highlighted here.

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

# A. Limitation

**Attention-Based Proxy Limitations.** Our framework relies on attention recall as a surrogate objective for compression quality, which empirically correlates with downstream task performance (Figure 2(a)). However, this correlation is not strictly monotonic across all scenarios. Prior work has noted that attention weights do not always faithfully reflect token importance for final predictions (Chen et al., 2024), and certain tasks may depend on tokens receiving low attention scores. Consequently, EntroKV is primarily compatible with compression operators that leverage attention scores for importance estimation (e.g., eviction and merging methods). Extending our entropy-guided allocation to attention-agnostic compression paradigms—such as learned importance predictors or activation-based pruning—remains an open direction.

**Scope of Evaluation.** Our experiments focus exclusively on text-based large language models (Llama-3.1 and Qwen2.5) and NLP benchmarks. The effectiveness of entropy-guided budget allocation in multimodal settings—where attention patterns may exhibit fundamentally different characteristics due to cross-modal interactions between vision and language tokens—has not been explored. Multimodal LLMs often process heterogeneous token sequences with varying information densities across modalities, potentially requiring modality-aware extensions to our allocation strategy.

**Systems-Level Optimization.** Although EntroKV introduces negligible computational overhead for entropy computation, we have not fully optimized the end-to-end inference pipeline. The current implementation computes head-wise budgets sequentially, whereas a fused CUDA kernel could parallelize entropy computation and budget allocation with the attention operator. Additionally, memory layout optimizations for variable-length KV caches across heads may further reduce memory fragmentation and improve cache efficiency. We leave such engineering optimizations for future work.

# B. Extended Experimental Results and Visualizations

## B.1. Layer–Head Level Recall–Budget Characteristics

Figure 7 illustrates the head-wise attention recall as a function of the Top-$K$ KV cache budget for two representative tasks, CWE and NIAH. The figure reveals substantial heterogeneity across layer–head pairs: some heads quickly saturate with a small budget, while others require significantly larger budgets to preserve attention mass. Moreover, NIAH generally exhibits slower recall saturation than CWE for many heads, indicating higher compression difficulty under long-context retrieval settings. These observations highlight that KV cache demand is highly non-uniform across heads and tasks, providing additional empirical motivation for fine-grained, entropy-guided budget allocation.

## B.2. Subtask Analysis in LongBench

Table 3 reports per-subtask results on LongBench under a fixed 30% KV retention budget. We analyze the gains brought by entropy-guided allocation from three perspectives: (i) QA with multi-hop retrieval, (ii) summarization with long-range content aggregation, and (iii) synthetic/few-shot tasks with heavy prompt redundancy.

*Table 3.* Detailed task performance on LongBench with 30% memory retention budget.

| | Method * | Single-Doc QA | | | Multi-Doc QA | | | Summarization | | | Few-shot Learning | | | Synthetic | | Code | | |
|---|---|---|---|---|---|---|---|---|---|---|---|---|---|---|---|---|---|---|
| | | NrtvQA | Qasper | MF-en | HotpotQA | 2WikiMQA | Musique | GovReport | QMSum | MultiNews | TREC | TriviaQA | SAMSum | PCount | PRe | Lcc | RB-P | AVG |
| **Llama** | Full Cache | 29.49 | 44.09 | 56.16 | 57.07 | 45.80 | 31.05 | 34.70 | 25.39 | 26.43 | 71.50 | 89.32 | 42.42 | 6.46 | 99.50 | 59.71 | 53.13 | 48.26 |
| | SnapKV | 29.05 | 42.53 | 54.85 | 56.10 | 46.19 | 30.70 | 30.60 | 24.77 | 23.44 | 69.50 | 87.95 | 41.10 | 5.33 | 99.00 | 58.19 | 52.93 | 47.01 |
| | Entro-SnapKV | 29.06 | 43.69 | 55.95 | 56.88 | 48.62 | 33.24 | 30.82 | 24.91 | 23.99 | 71.83 | 91.07 | 40.67 | 7.09 | 99.33 | 59.82 | 53.46 | 48.15 |
| | CaM | 29.90 | 41.51 | 54.66 | 56.28 | 49.40 | 31.10 | 30.35 | 24.98 | 23.65 | 70.00 | 88.93 | 40.23 | 6.25 | 99.50 | 58.78 | 51.38 | 47.31 |
| | Entro-CaM | 31.42 | 44.92 | 56.49 | 57.84 | 48.98 | 33.25 | 31.27 | 25.57 | 24.35 | 72.50 | 91.46 | 41.32 | 9.75 | 100.00 | 61.66 | 53.53 | 49.02 |
| | D2O | 29.86 | 42.75 | 55.32 | 57.10 | 47.60 | 31.09 | 30.64 | 24.82 | 23.66 | 69.67 | 90.89 | 41.49 | 7.28 | 99.50 | 60.39 | 53.00 | 47.82 |
| | Entro-D2O | 30.28 | 44.77 | 57.53 | 58.93 | 48.99 | 33.11 | 31.08 | 24.94 | 24.10 | 74.50 | 90.30 | 41.98 | 11.38 | 99.50 | 60.48 | 53.96 | 49.11 |
| **Qwen** | Full Cache | 28.50 | 43.42 | 52.82 | 58.93 | 45.21 | 30.70 | 31.51 | 23.55 | 23.36 | 72.00 | 89.21 | 44.98 | 10.50 | 100.00 | 57.31 | 63.50 | 48.47 |
| | SnapKV | 28.08 | 42.85 | 53.06 | 57.72 | 45.90 | 30.91 | 29.29 | 23.36 | 20.38 | 68.67 | 88.54 | 43.22 | 7.17 | 100.00 | 55.97 | 62.33 | 47.34 |
| | Entro-SnapKV | 28.58 | 43.20 | 53.41 | 59.13 | 47.93 | 30.05 | 29.46 | 23.63 | 20.96 | 70.50 | 88.46 | 45.36 | 9.50 | 100.00 | 56.73 | 63.17 | 48.13 |
| | CaM | 27.93 | 41.71 | 51.81 | 57.66 | 44.32 | 29.50 | 30.24 | 23.38 | 21.89 | 67.50 | 88.30 | 43.55 | 7.00 | 99.50 | 54.85 | 61.77 | 46.93 |
| | Entro-CaM | 29.54 | 44.71 | 53.39 | 59.00 | 46.67 | 31.84 | 30.53 | 24.12 | 22.34 | 71.50 | 89.85 | 45.35 | 9.50 | 100.00 | 57.61 | 64.35 | 48.77 |
| | D2O | 28.62 | 41.03 | 51.21 | 57.16 | 43.80 | 29.64 | 19.53 | 20.76 | 15.95 | 70.00 | 86.39 | 36.15 | 8.00 | 46.67 | 50.60 | 54.12 | 41.23 |
| | Entro-D2O | 29.59 | 43.44 | 53.85 | 57.48 | 45.71 | 29.92 | 24.09 | 22.16 | 17.68 | 70.25 | 89.98 | 35.33 | 9.5 | 99.5 | 51.85 | 53.91 | 45.89 |

**(1) Multi-document / multi-hop QA benefits the most.** For Llama-3.1, entropy-guided allocation consistently improves multi-hop QA tasks such as HotpotQA and Musique. Both tasks exhibit notable performance gains when switching from fixed-budget baselines to their entropy-guided counterparts. Similarly, retrieval-heavy TriviaQA shows substantial

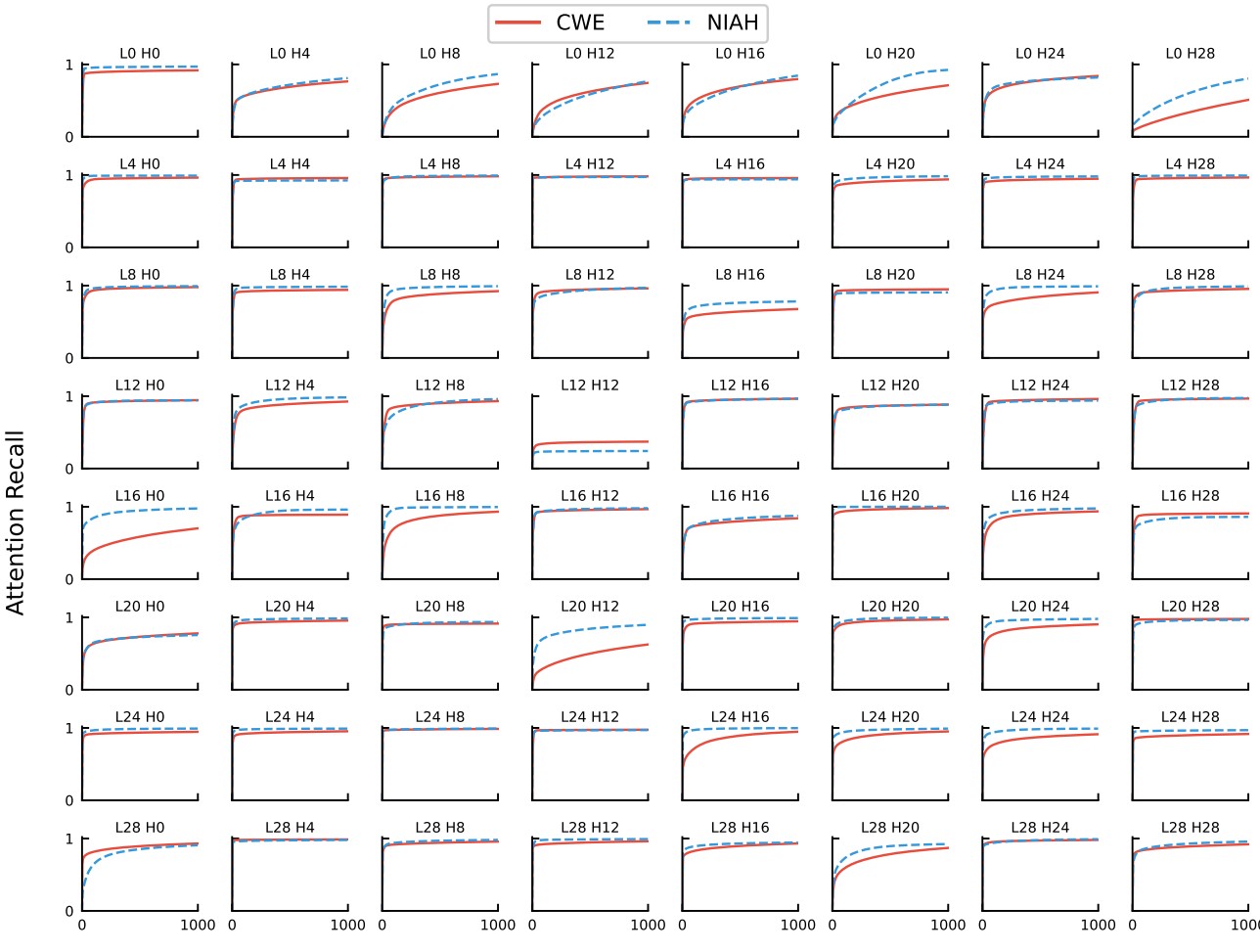

*Figure 7.* Head-wise attention recall as a function of the Top-$K$ KV cache budget for two representative tasks: Common Word Extraction (CWE) and Needle-In-A-Haystack (NIAH). Each subplot corresponds to a specific layer–head pair $(l, h)$. The curves reveal substantial heterogeneity across heads and layers, as well as clear task-dependent differences in compression difficulty, motivating entropy-guided, fine-grained KV cache budget allocation.

improvement, approaching Full Cache performance levels. This pattern suggests that these tasks induce more *diffuse* attention (higher entropy) across long contexts, where uniform (or static) budgets more frequently discard long-tail but critical evidence tokens.

For Qwen2.5, the improvements are also stable on multi-hop tasks, with consistent gains observed across HotpotQA and related benchmarks, indicating that entropy remains a robust scheduling signal across architectures.

**(2) Summarization tasks show moderate but stable gains.** On long-document summarization (GovReport, QMSum, MultiNews), entropy guidance generally improves or maintains performance relative to fixed-budget baselines. The gains are smaller than QA, which is consistent with the fact that summarization typically exhibits more localized "block-wise" salient spans (lower entropy in many heads), where aggressive compression in low-entropy heads has limited negative impact, while selectively preserving high-entropy heads still helps maintain cross-section coherence.

**(3) Few-shot / synthetic tasks sometimes exceed the Full Cache baseline.** On tasks with strong prompt redundancy or distractors (e.g., synthetic counting and pattern tasks), we observe cases where entropy-guided compressed inference approaches or slightly surpasses Full Cache. A plausible explanation is that entropy-guided allocation behaves as an *implicit denoiser*: it compresses low-entropy heads (often responsible for repetitive or template-like attention) more aggressively,

while allocating more budget to high-entropy heads that carry decision-relevant dependencies. This effect matches the qualitative observation in the main text that Full Cache does not always monotonically benefit from longer few-shot prompts.

**Summary.** Overall, Table 3 indicates that entropy-guided allocation yields the largest gains on tasks that require long-range evidence aggregation and multi-hop retrieval, while offering consistent improvements (or no regressions) on summarization and synthetic/few-shot tasks under the same memory budget.

## B.3. Subtask Analysis in Ruler

Tables 4 and 5 report subtask-level results on RULER at 4K and 16K context lengths, respectively, under a fixed 30% retention budget. RULER is specifically designed to stress-test long-context capabilities, making it particularly diagnostic for failure modes of KV cache compression.

*Table 4.* Detailed task performance on RULER-4K benchmark with 30% memory retention budget.

| | Method * | NIAH.S.1 | NIAH.S.2 | NIAH.S.3 | NIAH.MK.1 | NIAH.MK.2 | NIAH.MK.3 | NIAH.MV | NIAH.MQ | VT | CWE | FWE | QA.1 | QA.2 | AVG |
|---|---|---|---|---|---|---|---|---|---|---|---|---|---|---|---|
| **Llama** | Full Cache | 100.00 | 99.67 | 100.00 | 100.00 | 100.00 | 99.50 | 100.00 | 99.79 | 100.00 | 98.42 | 93.56 | 85.67 | 63.33 | 95.38 |
| | SnapKV | 100.00 | 99.67 | 66.33 | 100.00 | 100.00 | 34.33 | 100.00 | 99.67 | 99.70 | 87.35 | 89.83 | 85.17 | 61.83 | 86.45 |
| | Entro-SnapKV | 100.00 | 100.00 | 66.50 | 100.00 | 100.00 | 54.00 | 100.00 | 99.88 | 99.90 | 92.30 | 91.50 | 86.50 | 70.69 | 89.33 |
| | CaM | 100.00 | 99.00 | 39.50 | 100.00 | 99.00 | 32.50 | 100.00 | 99.88 | 98.80 | 85.95 | 89.50 | 84.50 | 59.50 | 83.70 |
| | Entro-CaM | 100.00 | 99.50 | 68.50 | 100.00 | 100.00 | 57.00 | 100.00 | 100.00 | 99.90 | 92.55 | 91.83 | 87.00 | 63.50 | 89.21 |
| | D2O | 99.80 | 99.30 | 55.80 | 100.00 | 99.50 | 42.40 | 99.72 | 99.65 | 98.98 | 83.61 | 88.60 | 87.10 | 62.00 | 85.88 |
| | Entro-D2O | 100.00 | 100.00 | 65.00 | 100.00 | 99.50 | 53.00 | 99.75 | 99.88 | 99.50 | 92.15 | 91.33 | 86.00 | 64.50 | 88.51 |
| **Qwen** | Full Cache | 100.00 | 100.00 | 100.00 | 100.00 | 100.00 | 99.50 | 99.46 | 100.00 | 99.83 | 99.23 | 88.78 | 83.33 | 59.83 | 94.61 |
| | SnapKV | 100.00 | 100.00 | 100.00 | 100.00 | 100.00 | 99.50 | 99.50 | 100.00 | 90.50 | 92.95 | 85.67 | 82.50 | 58.50 | 93.09 |
| | Entro-SnapKV | 100.00 | 100.00 | 100.00 | 100.00 | 100.00 | 100.00 | 99.62 | 100.00 | 92.20 | 91.50 | 86.67 | 84.00 | 60.00 | 93.38 |
| | CaM | 100.00 | 100.00 | 100.00 | 100.00 | 100.00 | 99.50 | 99.62 | 100.00 | 86.80 | 87.20 | 83.50 | 81.50 | 57.50 | 91.97 |
| | Entro-CaM | 100.00 | 100.00 | 100.00 | 100.00 | 100.00 | 99.60 | 99.70 | 100.00 | 99.42 | 97.61 | 87.07 | 83.30 | 59.70 | 94.34 |
| | D2O | 100.00 | 100.00 | 97.50 | 100.00 | 100.00 | 99.00 | 98.88 | 97.00 | 66.50 | 69.30 | 77.50 | 79.00 | 55.00 | 87.67 |
| | Entro-D2O | 100.00 | 100.00 | 97.00 | 100.00 | 100.00 | 99.00 | 99.75 | 98.50 | 88.20 | 80.90 | 82.67 | 80.00 | 58.00 | 91.08 |

*Table 5.* Detailed task performance on RULER-16K benchmark with 30% memory retention budget.

| | Method * | NIAH.S.1 | NIAH.S.2 | NIAH.S.3 | NIAH.MK.1 | NIAH.MK.2 | NIAH.MK.3 | NIAH.MV | NIAH.MQ | VT | CWE | FWE | QA.1 | QA.2 | AVG |
|---|---|---|---|---|---|---|---|---|---|---|---|---|---|---|---|
| **Llama** | Full Cache | 100.00 | 100.00 | 100.00 | 99.50 | 100.00 | 99.50 | 99.75 | 99.75 | 100.00 | 66.00 | 89.50 | 82.00 | 56.50 | 91.73 |
| | SnapKV | 100.00 | 100.00 | 100.00 | 99.50 | 100.00 | 89.00 | 99.62 | 99.62 | 99.40 | 70.85 | 90.00 | 79.50 | 54.00 | 90.88 |
| | Entro-SnapKV | 100.00 | 100.00 | 100.00 | 99.50 | 100.00 | 95.50 | 99.75 | 99.62 | 99.40 | 68.30 | 89.67 | 82.50 | 57.00 | 91.63 |
| | CaM | 100.00 | 99.50 | 11.00 | 98.00 | 58.00 | 2.00 | 97.25 | 99.38 | 84.10 | 27.75 | 64.83 | 79.00 | 52.50 | 67.18 |
| | Entro-CaM | 100.00 | 100.00 | 100.00 | 99.50 | 100.00 | 95.00 | 99.75 | 100.00 | 100.00 | 69.85 | 90.83 | 81.00 | 54.50 | 91.57 |
| | D2O | 99.50 | 98.00 | 26.50 | 99.50 | 98.50 | 3.50 | 98.38 | 99.50 | 96.70 | 27.95 | 89.67 | 79.00 | 54.50 | 74.71 |
| | Entro-D2O | 99.00 | 100.00 | 98.50 | 100.00 | 99.50 | 94.50 | 99.00 | 99.88 | 98.30 | 66.95 | 89.67 | 81.50 | 54.50 | 90.87 |
| **Qwen** | Full Cache | 100.00 | 100.00 | 99.67 | 100.00 | 99.50 | 93.00 | 95.62 | 100.00 | 98.43 | 86.12 | 91.89 | 65.33 | 51.17 | 90.83 |
| | SnapKV | 100.00 | 100.00 | 99.50 | 100.00 | 100.00 | 94.00 | 96.38 | 100.00 | 98.70 | 62.30 | 95.83 | 64.00 | 49.00 | 89.21 |
| | Entro-SnapKV | 100.00 | 100.00 | 99.00 | 100.00 | 100.00 | 94.50 | 96.88 | 100.00 | 98.90 | 58.85 | 96.17 | 66.00 | 51.50 | 89.37 |
| | CaM | 100.00 | 99.50 | 99.00 | 100.00 | 99.50 | 94.00 | 95.12 | 99.88 | 89.70 | 13.00 | 95.50 | 65.50 | 50.00 | 84.67 |
| | Entro-CaM | 100.00 | 100.00 | 100.00 | 100.00 | 100.00 | 95.00 | 96.75 | 100.00 | 98.90 | 72.35 | 96.33 | 65.50 | 52.00 | 90.53 |
| | D2O | 100.00 | 100.00 | 98.50 | 100.00 | 98.50 | 92.50 | 94.00 | 98.88 | 79.20 | 11.35 | 92.33 | 62.00 | 47.50 | 82.67 |
| | Entro-D2O | 100.00 | 100.00 | 96.00 | 100.00 | 99.50 | 92.50 | 95.88 | 98.25 | 86.00 | 55.05 | 87.17 | 63.50 | 49.50 | 86.41 |

**(1) NIAH family reveals catastrophic failures of static allocation.** Across both 4K and 16K, most NIAH subtasks (S1/S2, MK1/MK2, MV/MQ) remain near-saturated for many methods, but the harder variants (notably S3 and MK3) exhibit sharp drops under fixed-budget compression. For Llama-3.1 at 4K (Table 4), SnapKV experiences significant degradation on NIAH-S3 and NIAH-MK3 compared to Full Cache, while Entro-SnapKV substantially recovers performance on these challenging subtasks. Similar "cliff" behaviors are even more pronounced for merging-based operators, where EntroKV dramatically recovers performance by preventing over-merging on high-entropy heads.

At 16K (Table 5), the failure mode becomes more severe for merging operators on Llama-3.1: CaM suffers near-complete collapse on NIAH-S3 and NIAH-MK3, while Entro-CaM restores performance to levels approaching Full Cache. D2O exhibits similar catastrophic degradation on these hard variants, while Entro-D2O successfully recovers the lost performance. These results strongly support the central motivation that uniform budgets cause severe mismatch between component-wise compression difficulty and allocated capacity, leading to catastrophic information loss on diffusion-heavy heads.

**(2) CWE/FWE highlight global-statistics sensitivity at long contexts.** CWE and FWE require retaining globally distributed evidence across the prompt. At 16K, we observe that merging-based baselines can severely underperform on CWE, while entropy-guided variants substantially recover performance (Table 5). This aligns with the entropy–recall monotonicity: tasks whose attention is intrinsically more uniform (high entropy) need larger budgets to preserve sufficient Top-$K$ mass.

**(3) QA1/QA2 exhibit consistent improvements but smaller headroom.** QA subtasks generally improve with entropy-guided allocation across both 4K and 16K, but the magnitude is smaller than NIAH hard variants. This is expected because QA tasks often have more localized evidence spans than NIAH "needle" settings, yielding fewer extreme high-entropy heads. Nevertheless, EntroKV still improves robustness by selectively protecting diffusion-heavy heads that mediate long-range dependency routing.

**Summary.** RULER subtask results (Tables 4 and 5) show that entropy-guided allocation mainly reduces *catastrophic* failures under long contexts, especially for hard NIAH variants and merging-based operators. This indicates that entropy is an effective online proxy for compression sensitivity, enabling near-oracle head-wise budget scheduling under strict global memory constraints.

## C. Experimental Setup and Reproducibility Details

### C.1. Prompt Templates for All Benchmarks

**LongBench and RULER.** For **LongBench** and **RULER**, we strictly follow the *default prompt templates* provided in the original benchmark papers / official implementations, without any task-specific rewriting or manual prompt tuning (Bai et al., 2023; Hsieh et al., 2024). This design choice is made to avoid irreproducibility caused by hidden prompt engineering.

**MATH-500 and SQuAD.** For short-context benchmarks, we use unified chat-style templates (system/user/assistant) with optional $k$-shot demonstrations. All few-shot examples are sampled independently from the evaluation set (disjoint from test instances).

---

**MATH-500 Prompt Template (chat format, optional $k$-shot)**

**System:** You are a math expert. Solve the following problem. Provide very concise reasoning, then output only the final answer inside \boxed{answer}. Do not include anything after the boxed answer.

**Demonstrations (optional, repeated for each example $i = 1..k$):**

> **User:** Question: {example_problem$_i$}
> **Assistant:** Answer: {example_solution$_i$}

**Query:**

> **User:** Question: {problem}
> **Assistant:** {model_generation}

---

**SQuAD Prompt Template (chat format, optional $k$-shot)**

**System:** You are a helpful assistant. Answer the question using the given context. Return a short answer span from the context.

**Demonstrations (optional, repeated for each example $i = 1..k$):**

> **User:** Context: {example_context$_i$}
> Question: {example_question$_i$}
> **Assistant:** Answer: {example_answer_span$_i$}

**Query:**

> **User:** Context: {context}
> Question: {question}
> **Assistant:** {model_generation}

---

### C.2. Inference Hyperparameters (Sampling & Decoding)

Table 6 summarizes the decoding and sampling hyperparameters used in all experiments. The same configuration is applied across all benchmarks (LongBench, RULER, MATH-500, and SQuAD), all backbone models, and all evaluated methods, including Full Cache, SnapKV, CaM, D2O, and EntroKV variants.

*Table 6.* Unified decoding and sampling hyperparameters used across all experiments.

| Hyperparameter | Value |
|---|---|
| Temperature ($T$) | 0.6 |
| Top-$p$ (nucleus sampling) | 0.9 |
| Top-$k$ | None |
| Repetition penalty | 1.0 |

Table 7 reports the SnapKV-style base compression and D2O/CaM merging hyperparameters used for all backbone models.

*Table 7.* Summary of key KV-cache compression configuration parameters used for each backbone model.

| Backbone | Base KV Compression (SnapKV-style) | | | | Merge (D2O / CaM) | | |
|---|---|---|---|---|---|---|---|
| | window_size | pooling | pooling_kernel | gqa_func | d2o_ema_beta | cam_merge_alpha | cam_use_bernoulli |
| **Llama-3.1-8B-Instruct** | 32 | maxpool | 7 | mean | 0.5 | 1.0 | True |
| **Qwen2.5-7B-Instruct** | 32 | maxpool | 75 | mean | 0.5 | 1.0 | True |

# D. Calibration Set and Global Baseline Entropy

This appendix describes the computation of the global baseline entropy $\bar{H}$, which serves as a stable reference for entropy-guided budget allocation in EntroKV.

### D.1. Calibration Set Construction

To ensure that the global baseline $\bar{H}$ captures diverse attention patterns across various task types and input characteristics, we construct the calibration set using a representative subset of **LongBench** (Bai et al., 2023). LongBench encompasses a wide range of long-context tasks, including:

- **Single-Document QA:** NarrativeQA, Qasper, MultiFieldQA-en

- **Multi-Document QA:** HotpotQA, 2WikiMultihopQA, MuSiQue

- **Summarization:** GovReport, QMSum, MultiNews

- **Few-shot Learning:** TREC, TriviaQA, SAMSum

- **Synthetic Tasks:** PassageCount, PassageRetrieval

- **Code Completion:** LCC, RepoBench-P

This diversity ensures that $\bar{H}$ reflects a broad spectrum of attention behaviors, ranging from highly localized patterns (low entropy) in retrieval-focused tasks to more diffuse patterns (high entropy) in summarization and multi-hop reasoning tasks.

### D.2. Computation Procedure

For each backbone model, we compute $\bar{H}$ as follows:

1. **Sampling:** We randomly sample $N = 500$ instances from the LongBench calibration split, stratified across task categories to ensure balanced coverage.

2. **Entropy Extraction:** For each sampled instance $t \in [N]$, we perform a forward pass and extract the normalized attention entropy $\tilde{H}_l^h(t)$ for all layer–head pairs $(l, h)$ at the observation window (following the same windowing strategy as the compression operator).

3. **Aggregation:** The global baseline is computed as the average normalized entropy across all instances, layers, and heads:

$$\bar{H} = \frac{1}{N} \sum_{t=1}^{N} \frac{1}{L \cdot H} \sum_{l=1}^{L} \sum_{h=1}^{H} \tilde{H}_l^h(t). \tag{7}$$

### D.3. Calibration Results

Table 8 reports the computed global baseline entropy $\bar{H}$ for each backbone model used in our experiments.

*Table 8.* Global baseline entropy $\bar{H}$ computed on the LongBench calibration set for each backbone model.

| BACKBONE MODEL | $\bar{H}$ |
|---|---|
| LLAMA-3.1-8B-INSTRUCT | 0.30 |
| QWEN2.5-7B-INSTRUCT | 0.45 |
| QWEN3-8B | 0.80 |

### D.4. Discussion

The observed difference in $\bar{H}$ between Llama-3.1-8B-Instruct (0.30) and Qwen2.5-7B-Instruct (0.45) reflects inherent architectural and training differences between these models:

- **Attention Pattern Characteristics:** Qwen2.5-7B-Instruct exhibits more diffuse attention distributions on average, potentially due to differences in pre-training data composition, attention mechanism design (e.g., different RoPE configurations), or model capacity allocation across layers.

- **Implications for Allocation:** A higher $\bar{H}$ for Qwen2.5 implies that, on average, its attention heads are harder to compress under uniform budgets. EntroKV automatically adapts to this by using the model-specific $\bar{H}$ as the baseline, ensuring that the allocation rule remains well-calibrated regardless of the underlying attention characteristics.

**Practical Note.** The calibration procedure is performed *once* per backbone model and does not require re-computation for different downstream tasks or deployment scenarios. The computed $\bar{H}$ values are stored as model-specific constants and loaded at inference time, incurring no additional runtime overhead.

## E. Integration with KV Cache Compression Operators

EntroKV is designed as a lightweight *front-end allocator* that decouples **budget scheduling** (how much to keep) from **compression execution** (how to compress). Given any backend KV compression operator, EntroKV only needs (i) access to the attention weights already produced during prefill/decoding and (ii) a control knob that specifies the target retention (e.g., top-$K$ length or merge ratio). In this section, we detail how EntroKV integrates with a representative eviction-based operator, **SnapKV**, without modifying its core mechanism.

### E.1. Adapting EntroKV to SnapKV

**SnapKV recap (backend).** SnapKV performs selective forgetting in a windowed manner. At each compression step, it uses an *observation window* of recent queries (window size $w$) to estimate token importance for cached keys/values, typically via pooling over attention scores (e.g., max-pooling with a kernel), and then retains a fixed quota of important KV entries (e.g., top-$K$) together with a small set of always-kept tokens (e.g., sink tokens and/or recent tokens). This design makes SnapKV efficient and compatible with chunk-wise inference.

**Key idea: replace fixed quota with entropy-guided head-wise quota.** The original SnapKV uses a *static* retention quota that is uniform across layers and heads. EntroKV keeps SnapKV's scoring, pooling, and token selection rules *unchanged*, and only replaces the *fixed* top-$K$ quota by a *dynamic head-wise* quota $K_l^h$ computed from attention entropy:

$$K_l^h = \left\lfloor m \cdot B_l^h \right\rfloor, \qquad B_l^h = \beta \left[ (1 - \alpha)\bar{H} + \alpha\tilde{H}_l^h \right], \tag{8}$$

where $m$ is the number of candidate KV entries (or the effective memory length) at the current step, $\tilde{H}_l^h \in [0, 1]$ is the normalized attention entropy for head $(l, h)$, $\bar{H}$ is the calibrated global baseline entropy, and $\alpha, \beta$ control adaptivity and total budget.

**Entropy computation aligned with SnapKV's observation window.** To ensure that the scheduling signal is *online*, *local*, and consistent with SnapKV, we compute entropy using the same observation-window protocol: we treat the last $w$ tokens as the query window, and measure the dispersion of their attention over the preceding cached positions. Concretely, for head $(l, h)$, let $A^{(l,h)} \in \mathbb{R}^{w \times n}$ denote the attention weights of the last $w$ queries attending to $n$ cached keys. We compute:

$$H\left(A^{(l,h)}\right) = -\frac{1}{w} \sum_{i=1}^{w} \sum_{j=1}^{n} A_{ij}^{(l,h)} \log A_{ij}^{(l,h)}, \qquad \tilde{H}_l^h = \frac{H(A^{(l,h)})}{\log n}. \tag{9}$$

This choice has two practical advantages: (i) **no extra matmul** is introduced since $A^{(l,h)}$ is already computed by the attention kernel; (ii) the signal is naturally synchronized with SnapKV's compression granularity (once per window).

**Budget conservation and discretization.** Since SnapKV requires integer quotas, we discretize $B_l^h$ into $K_l^h$. In implementation, we optionally apply a per-step normalization to strictly match the desired global retention ratio (especially when $n$ varies across steps):

$$\hat{B}_l^h = \beta \cdot \frac{B_l^h}{\frac{1}{LH} \sum_{l',h'} B_{l'}^{h'}}, \qquad K_l^h = \lfloor m \cdot \hat{B}_l^h \rfloor, \tag{10}$$

and then adjust a small remainder to ensure $\sum_{l,h} K_l^h$ matches the target exactly. This keeps the total memory footprint identical to the fixed-budget SnapKV baseline, while redistributing capacity toward high-entropy heads.

**Plug-and-play implementation: what changes and what stays unchanged.** Our adaptation is deliberately minimal:

- **Unchanged (SnapKV core):** observation window, importance score computation, pooling type (e.g., maxpool), pooling kernel size, and the original retention policy (top-$K$ + always-keep set). In our experiments we use the same SnapKV hyperparameters as the baseline (e.g., `window_size`, `pooling`, `pooling_kernel`, and the GQA aggregation function `gqa_func`).

- **Changed (only quota):** replace the uniform quota $K$ by the entropy-guided $K_l^h$ for each head.

**Handling GQA / multi-query attention.** For backbones using grouped-query attention (GQA), multiple query heads may share the same KV head. We compute $\tilde{H}_l^h$ at the query-head granularity and then map it to KV-head quotas by aggregation (e.g., averaging entropies of query heads that share a KV head, consistent with `gqa_func` used in the backend). This preserves SnapKV's KV sharing semantics while enabling head-wise scheduling.

**Algorithmic pipeline (Entro-SnapKV).** At each windowed compression step:

1. **Reuse attention:** obtain attention weights for the last $w$ queries from the standard attention computation.

2. **Compute entropy:** compute $\tilde{H}_l^h$ for every layer-head pair using the windowed attention matrix.

3. **Allocate quota:** compute head-wise budgets $B_l^h$ and discretize to $K_l^h$ (optionally normalized to match the global ratio).

4. **Run SnapKV:** feed $\{K_l^h\}$ into SnapKV and execute its standard pooling + top-$K$ retention (plus always-keep tokens).

**Complexity and systems compatibility.** Entropy computation is a simple reduction over existing attention weights ($\sum a \log a$), with per-step overhead $O(L \cdot H \cdot w \cdot n)$ on the already materialized windowed attention, which is negligible compared to attention kernel cost in practice. Since we do not alter the attention operator nor introduce new attention patterns, Entro-SnapKV remains compatible with FlashAttention-style kernels and chunk-wise prefill/decoding.

**Takeaway.** EntroKV turns SnapKV from a *fixed-quota* eviction strategy into an *entropy-adaptive* scheduler: high-entropy (diffuse, compression-sensitive) heads automatically receive larger quotas, while low-entropy heads are more aggressively compressed, improving accuracy at the same total KV budget.

### E.2. Adapting EntroKV to Merging Method

Beyond SnapKV, EntroKV naturally generalizes to other KV compression methods that operate on a fixed *observation window* to maintain compatibility with FlashAttention-style kernels. We briefly discuss two representative cases.

**CaM (Cache Merging).** CaM (Zhang et al., 2024) performs value-state merging based on token similarity, where candidate tokens are selected from a recent observation window and then merged into preserved KV entries under a fixed merging ratio. EntroKV integrates with CaM by treating the merging ratio as a *budget variable* rather than a static hyperparameter. Specifically, we reuse the same windowed attention weights used by CaM to compute head-wise attention entropy, and dynamically allocate merging budgets (e.g., the number of tokens to be merged) according to $K_l^h$. The similarity computation, matching, and weighted merging rules of CaM remain unchanged; EntroKV only modulates how aggressively each head performs merging. As CaM already restricts operations within a local window, this adaptation preserves full FlashAttention compatibility.

**D2O (Dynamic Discriminative Operations).** D2O (Wan et al., 2025) adopts an observation-window-based eviction strategy and explicitly maintains recent tokens and attention sink tokens to support efficient long-context inference. EntroKV complements D2O at the scheduling level by replacing its uniform or layer-level cache allocation with a finer-grained entropy-guided head-wise allocation. Entropy is computed from the same windowed attention maps already used by D2O, introducing no additional attention computation. Both the token eviction mechanism and the EMA-based dynamic token merging in D2O are left intact; EntroKV only determines the effective cache size (or eviction quota) per head. Since all decisions are derived from local attention windows, the combined EntroKV–D2O pipeline remains fully compatible with FlashAttention and streaming decoding.

## F. Optimization Derivation for Entropy-Guided Budget Allocation

This appendix provides an optimization-oriented derivation of our entropy-guided KV budget allocation rule, grounded in the paper's resource allocation objective and the attention-recall surrogate.

### F.1. Problem Setup: Resource Allocation Objective

Consider a Transformer with $L$ layers and $H$ attention heads per layer. Let $B_l^h \geq 0$ denote the retention budget assigned to head $(l, h)$, under a global budget constraint $\sum_{l,h} B_l^h \leq B$. The paper formulates KV cache compression as a constrained resource allocation problem:

$$\min_{\{B_l^h\}} \sum_{l=1}^{L} \sum_{h=1}^{H} \mathcal{L}_l^h(B_l^h) \quad \text{s.t.} \quad \sum_{l,h} B_l^h \leq B, \tag{11}$$

where $\mathcal{L}_l^h(\cdot)$ denotes the head-wise compression loss under budget $B_l^h$.

### F.2. Surrogate Objective: Maximizing Attention Recall

Directly optimizing $\mathcal{L}_l^h(B_l^h)$ during inference is challenging since it is (i) not available a priori, and (ii) typically non-differentiable w.r.t. discrete budget decisions. Following the main text, we adopt *attention recall* as an actionable surrogate. For Top-$K$ style retention (or an equivalent budgeted compression operator), define the head-wise recall as the retained attention mass:

$$R_l^h(B) \triangleq \sum_{i \in \text{Top-}B} a_i^{(l,h)}, \tag{12}$$

where $a^{(l,h)} \in \Delta^{n-1}$ is the attention distribution over $n$ context positions and Top-$B$ denotes the indices of the largest $B$ attention weights.

The paper reports that higher recall strongly correlates with lower downstream performance loss. Accordingly, we consider the surrogate allocation problem:

$$\max_{\{B_l^h\}} \sum_{l,h} R_l^h(B_l^h) \quad \text{s.t.} \quad \sum_{l,h} B_l^h \leq B. \tag{13}$$

This surrogate captures the intuition that KV cache compression is accurate when it preserves most of the attention mass.

### F.3. Optimality Condition: Equalizing Marginal Recall Utility

Treating $B_l^h$ as continuous for analysis (later discretized), the Lagrangian of (13) is:

$$\mathcal{J}(\{B_l^h\}, \lambda) = \sum_{l,h} R_l^h(B_l^h) - \lambda\Big(\sum_{l,h} B_l^h - B\Big). \tag{14}$$

The Karush–Kuhn–Tucker (KKT) condition implies that for any head receiving positive budget ($B_l^h > 0$), the optimal solution satisfies:

$$\frac{\partial R_l^h(B_l^h)}{\partial B_l^h} = \lambda, \quad \forall(l,h) \text{ s.t. } B_l^h > 0, \tag{15}$$

i.e., the optimal allocation equalizes the *marginal recall gain per additional unit of budget* across heads. This formalizes why uniform allocation is suboptimal under heterogeneous head-wise recall–budget curves.

### F.4. From Recall to Entropy: Entropy as a Proxy for Compression Difficulty

While (15) characterizes the optimum, it is not directly actionable because $R_l^h(B)$ (and its derivative) is an a posteriori quantity. We therefore require an online-computable proxy that predicts head-wise *compression difficulty*.

**Normalized attention entropy.** We adopt the normalized entropy used in the main text:

$$\tilde{H}_l^h \triangleq \frac{H(a^{(l,h)})}{\log n} \in [0,1], \qquad H(a) = -\sum_{i=1}^n a_i \log a_i. \tag{16}$$

**Entropy–recall monotonicity (fixed budget).** Entropy quantifies dispersion of $a^{(l,h)}$: low entropy indicates a "peaky" distribution where Top-$B$ captures most mass, while high entropy indicates a diffuse distribution with a heavier tail. This motivates the following monotonic relationship.

**Lemma F.1** (Entropy correlates with compression difficulty). *Fix a budget $B$. For two attention distributions $a$ and $b$ over $n$ positions, if $a$ is more concentrated than $b$ (equivalently, $b$ is closer to uniform), then $H(a) \leq H(b)$ and the Top-$B$ mass satisfies*

$$\sum_{i \in \text{Top-}B} a_i \geq \sum_{i \in \text{Top-}B} b_i, \tag{17}$$

*i.e., higher entropy implies lower recall under the same budget.*

*Proof sketch.* The Shannon entropy is Schur-concave, increasing as a distribution becomes more uniform, while the Top-$B$ cumulative mass is Schur-convex, decreasing as mass spreads into the tail. Thus, increasing dispersion (higher entropy) reduces the achievable Top-$B$ captured mass under a fixed $B$. $\square$

**Difficulty as "budget to reach a target recall."** Define the minimal budget required to achieve a recall threshold $r \in (0,1)$:

$$D_l^h(r) \triangleq \min\{B : R_l^h(B) \geq r\}. \tag{18}$$

By Lemma F.1, heads with larger $\tilde{H}_l^h$ require larger $D_l^h(r)$ to reach the same $r$. Therefore, a principled allocation should be *monotonically increasing* in $\tilde{H}_l^h$:

$$B_l^h \propto g(\tilde{H}_l^h), \qquad g(\cdot) \text{ monotone increasing.} \tag{19}$$

### F.5. Practical Allocation Rule: Calibrated Linear Approximation

The remaining design choice is the functional form $g(\cdot)$. In principle, one could learn or fit $g$ from head-wise recall curves, but we seek a lightweight, training-free rule. Since $\tilde{H}_l^h \in [0, 1]$ is already normalized and comparable across inputs, we adopt a first-order (linear) approximation:

$$g(\tilde{H}_l^h) \approx c_0 + c_1 \tilde{H}_l^h, \quad c_1 > 0. \tag{20}$$

**Global calibration for stability.** Instantaneous entropy can be noisy. To stabilize allocation, we introduce a global baseline $\bar{H}$ computed on a calibration set (as in the main text), and interpolate between the baseline and the instance-specific entropy using $\alpha \in [0, 1]$:

$$\hat{H}_l^h \triangleq (1 - \alpha)\bar{H} + \alpha\tilde{H}_l^h. \tag{21}$$

**Scaling to match the total budget.** Finally, we use a global scaling factor $\beta > 0$ to control the overall retention ratio, yielding the implemented allocation rule:

$$B_l^h = \beta \cdot \hat{H}_l^h = \beta\Big((1 - \alpha)\bar{H} + \alpha\tilde{H}_l^h\Big). \tag{22}$$

This rule is consistent with the optimization principle in (15): since higher entropy heads have lower marginal recall under fixed budgets, allocating more budget to high-entropy heads compensates for their greater compression difficulty, improving the global recall under a fixed resource constraint.

**Interpretation of $\alpha$ and $\beta$.** $\alpha$ controls the bias–variance trade-off between a stable global prior ($\bar{H}$) and adaptive instance signals ($\tilde{H}_l^h$), while $\beta$ scales the overall budget to satisfy the desired retention ratio.

## G. Comparison of Attention-Distribution-Based Budget Allocation Signals

This section compares three attention-distribution-based signals for *dynamic KV cache budget allocation*. All definitions and normalization schemes are presented in a purely mathematical form, independent of any specific implementation, while remaining consistent with the allocation logic used in our experiments. We analyze why attention entropy provides a more stable and expressive allocation signal than L2-norm-based and TopK-based alternatives.

### G.1. Design Requirements for Allocation Signals

In the context of dynamic KV cache allocation, an allocation signal should satisfy the following requirements:

1. **Online computability**: the signal must be computable during inference without offline profiling or additional supervision;

2. **Operator independence**: it should depend only on the attention distribution, not on a specific compression operator;

3. **Normalized scale**: the signal should admit normalization to a comparable range (e.g., $[0, 1]$) across different sequence lengths;

4. **Consistent scheduling direction**: attention distributions that are more diffuse or harder to compress should receive larger budgets.

Under these principles, we compare three allocation signals: entropy-based allocation, L2-norm-based allocation, and coverage-based TopK allocation.

### G.2. Problem Setup and Notation

Let

$$\mathbf{p} = (p_1, p_2, \ldots, p_n), \quad \sum_{i=1}^{n} p_i = 1,$$

denote the attention distribution of a given attention head over $n$ cached tokens.

The goal of budget allocation is to assign a larger retention budget to distributions that are intrinsically harder to approximate under compression, while allocating fewer resources to distributions that are easy to compress.

### G.3. Entropy-Based Allocation

We use Shannon entropy to quantify the dispersion of the attention distribution:

$$H(\mathbf{p}) = -\sum_{i=1}^{n} p_i \log p_i.$$

Since the maximum entropy grows with sequence length ($H_{\max} = \log n$), we adopt a length-normalized form:

$$\tilde{H}(\mathbf{p}) = \frac{H(\mathbf{p})}{\log n} \in [0, 1].$$

A larger normalized entropy indicates a more uniform or long-tailed attention distribution, which is generally harder to approximate using a small retained subset. Accordingly, entropy-based allocation assigns larger budgets to heads with larger $\tilde{H}$, and smaller budgets to heads with more concentrated attention.

### G.4. L2-Norm-Based Allocation

An alternative measure of concentration is the $\ell_2$ norm of the attention distribution:

$$\|\mathbf{p}\|_2 = \left(\sum_{i=1}^{n} p_i^2\right)^{1/2},$$

which satisfies

$$\|\mathbf{p}\|_2 \in \left[\sqrt{\tfrac{1}{n}}, 1\right].$$

To obtain a length-invariant quantity, we apply linear normalization:

$$\tilde{N}(\mathbf{p}) = \frac{\|\mathbf{p}\|_2 - \sqrt{1/n}}{1 - \sqrt{1/n}} \in [0, 1].$$

Since larger $\ell_2$ norms correspond to sharper (and thus easier-to-compress) distributions, the allocation signal is defined by inversion:

$$D_{\mathrm{L2}}(\mathbf{p}) = 1 - \tilde{N}(\mathbf{p}).$$

Under this definition, more diffuse attention distributions receive larger allocation values, making the scheduling direction consistent with entropy-based allocation.

### G.5. Coverage-Based TopK Allocation (Effective-K)

The TopK signal considered in this work does not correspond to Recall@K. Instead, it measures the minimum number of tokens required to cover a fixed fraction of the total attention mass.

Given a coverage threshold $\gamma \in (0, 1)$, define

$$K^*(\gamma) = \min\left\{K \ \middle|\ \sum_{i=1}^{K} p_{(i)} \geq \gamma\right\},$$

where $p_{(i)}$ denotes the attention weights sorted in descending order.

We further normalize this quantity by sequence length:

$$\tilde{K}(\gamma) = \frac{K^*(\gamma)}{n} \in (0, 1].$$

A small $\tilde{K}$ indicates that most attention mass is concentrated on a few tokens, while a large $\tilde{K}$ implies that attention mass is widely distributed. Thus, $\tilde{K}$ can be interpreted as an explicit estimate of the *effective number of tokens* required to preserve attention mass under compression. However, its numerical value depends explicitly on the choice of coverage threshold $\gamma$.

### G.6. Comparison and Alignment of Allocation Signals

All three signals are aligned to the same scheduling direction: more diffuse attention distributions correspond to larger allocation values and thus larger budgets.

Despite this alignment, the signals differ in stability and expressive power. The L2 norm is dominated by high-probability tokens and is less sensitive to long-tail mass. The coverage-based TopK metric directly reflects how many tokens are required to reach a target coverage, but introduces an additional hyperparameter $\gamma$. Entropy, by contrast, provides a smooth, global characterization of the entire distribution and requires no extra thresholds.

### G.7. Empirical Implications

Under identical global KV cache budget constraints, we empirically compare entropy-based, L2-norm-based, coverage-based TopK, and fixed allocation strategies. Table 9 reports detailed results on the LongBench benchmark, showing that entropy-based allocation consistently achieves the best average performance across tasks.

*Table 9.* LongBench performance under different KV cache budget allocation signals on LLaMA-3.1-8B-Instruct.

| | Method * | Single-Doc QA | | | Multi-Doc QA | | | Summarization | | | Few-shot Learning | | | Synthetic | | Code | | |
|---|---|---|---|---|---|---|---|---|---|---|---|---|---|---|---|---|---|---|
| | | NrtvQA | Qasper | MF-en | HotpotQA | 2WikiMQA | Musique | GovReport | QMSum | MultiNews | TREC | TriviaQA | SAMSum | PCount | PRe | Lcc | RB-P | AVG |
| LongBench | Entropy | 29.06 | 43.69 | 55.95 | 56.88 | 48.62 | 33.24 | 30.82 | 24.91 | 23.99 | 71.83 | 91.07 | 40.67 | 7.09 | 99.33 | 59.82 | 53.46 | 48.15 |
| | topk | 28.54 | 41.60 | 54.51 | 58.93 | 48.38 | 31.78 | 26.53 | 24.38 | 23.89 | 71.00 | 88.19 | 42.09 | 7.67 | 99.00 | 59.73 | 53.23 | 47.47 |
| | L2Norm | 28.51 | 43.57 | 55.66 | 56.63 | 47.83 | 31.80 | 30.56 | 24.95 | 23.34 | 71.00 | 90.75 | 41.21 | 6.10 | 100.00 | 60.68 | 51.70 | 47.77 |
| | fixed | 29.05 | 42.53 | 54.85 | 56.10 | 46.19 | 30.70 | 30.60 | 24.77 | 23.44 | 69.50 | 87.95 | 41.10 | 5.33 | 99.00 | 58.19 | 52.93 | 47.01 |

### G.8. Summary

In summary, although entropy-based, L2-norm-based, and coverage-based TopK signals can all be normalized to support dynamic budget allocation, attention entropy offers a more principled and robust proxy for compression difficulty. By capturing the global structure of attention distributions without introducing additional hyperparameters, entropy is particularly well suited for dynamic KV cache budget allocation.

## H. Method-Level Comparison of Dynamic Budget Allocation Approaches

To further contextualize why entropy serves as a stronger proxy signal for dynamic KV cache budget allocation, we provide a detailed method-level comparison across proxy signal type, allocation granularity, and extra computational operations. Table 10 summarizes the key differences.

*Table 10.* Method-level comparison of dynamic budget allocation approaches. EntroKV achieves the finest allocation granularity (head + layer + task) with minimal computational overhead.

| METHOD | PROXY SIGNAL | HEAD | LAYER | TASK | EXTRA ALLOCATION OPERATIONS |
|---|---|---|---|---|---|
| PYRAMIDKV | LAYER-DEPTH PRIOR | ✗ | ✓ | ✗ | FIXED DEPTH-BASED LAYER SCHEDULE LOOKUP |
| ADAKV | ACCUMULATED ATTENTION MASS | ✓ | ✗ | ✗ | PER-HEAD ATTENTION-MASS ACCUMULATION AND SELECTION |
| LAVA | OUTPUT-LOSS SURROGATE | ✓ | ✓ | ✗ | RECENT-ATTENTION/VALUE-NORM STATISTICS AND RANKING |
| TASK-KV | SEMANTIC DIFFERENTIATION | ✓ | ✗ | ✓ | SEMANTIC-CENTER DISTANCE COMPUTATION AND HEAD PARTITIONING |
| CAKE | ENTROPY + TEMPORAL VARIANCE | ✗ | ✓ | ✗ | LAYER ENTROPY/VARIANCE STATISTICS AND CASCADING REDISTRIBUTION |
| **ENTROKV** | **NORMALIZED ATTENTION ENTROPY** | ✓ | ✓ | ✓ | **SINGLE ENTROPY REDUCTION OVER EXISTING ATTENTION WEIGHTS** |

**Key advantages of EntroKV over prior allocation methods.**

- **Length-normalized and directly comparable.** EntroKV uses normalized attention entropy $\tilde{H}_l^h = H(\mathbf{a}_l^h)/\log n \in [0,1]$, which makes allocation scores directly comparable across heads, layers, and inputs of varying lengths. In contrast, methods such as AdaKV (accumulated attention mass) and LAVa (output-loss surrogate) produce scores whose scales depend on sequence length or layer depth, requiring additional normalization heuristics.

- **No auxiliary computation branch at inference.** EntroKV reuses attention weights already materialized during the standard forward pass (e.g., within the SnapKV observation window), introducing only a lightweight entropy reduction ($\sum a_i \log a_i$) with no additional matrix multiplications. By comparison, LAVa requires computing recent-attention/value-norm statistics, Task-KV needs semantic-center distance computation, and CAKE requires both entropy and temporal variance statistics with cascading redistribution logic.

- **Finest allocation granularity.** EntroKV is the only method that simultaneously achieves head-level, layer-level, and task-level adaptive allocation through a single unified signal. PyramidKV and CAKE operate only at the layer level, AdaKV and Task-KV lack layer-level awareness, and LAVa does not adapt to task-level semantics. EntroKV's normalized entropy naturally captures all three dimensions: different heads within the same layer exhibit distinct entropy values (head-level), layer-averaged entropy varies non-monotonically across depth (layer-level), and the overall entropy distribution shifts with task characteristics (task-level), as demonstrated in Figure 6.

- **Theoretically motivated.** Under the assumptions in Section 3.2, entropy is Schur-concave and monotonically aligned with Top-$K$ recall difficulty (Appendix F.4), providing a principled link between attention dispersion and allocation demand. This distinguishes EntroKV from purely heuristic designs that lack formal justification for their proxy choice.

## I. Observation Window Length Ablation

We investigate the sensitivity of EntroKV to the observation window size $w$ used for entropy computation. Table 11 reports LongBench performance on Llama-3.1-8B-Instruct (Entro-SnapKV, retention ratio 0.3) across different window lengths.

*Table 11.* Effect of observation window size $w$ on LongBench performance (Llama-3.1-8B-Instruct, Entro-SnapKV).

| $w$ | 128 | 64 | 32 | 16 | 8 |
|---|---|---|---|---|---|
| LONGBENCH | 47.98 | 47.92 | 48.03 | 47.98 | 47.87 |

The results demonstrate remarkable stability: performance varies by less than 0.2 points across a $16\times$ range of window sizes (from $w = 8$ to $w = 128$). Even with $w = 8$ (only 8 recent queries), the entropy signal remains a reliable proxy for head-wise compression sensitivity. This robustness arises because attention entropy captures an *intrinsic structural property* of each head—namely, the degree of attention dispersion—which is largely invariant to the specific query subset used for estimation. The default setting $w = 32$ (matching SnapKV's standard observation window) provides an optimal balance between signal quality and computational overhead.

## J. Compatibility with KV Cache Quantization

EntroKV operates at the *budget level* (governing how many tokens to retain or merge), which is mechanistically orthogonal to *quantization* methods that operate at the *representation level* (reducing numerical precision of stored KV entries). To empirically verify this orthogonality, we evaluate EntroKV combined with standard INT4 KV-cache quantization on Llama-3.1-8B-Instruct.

### J.1. Joint Compression with Uniform Quantization

Table 12 reports the performance of combining Entro-SnapKV with 4-bit KV-cache quantization, compared to each method applied independently.

The combined configuration achieves an effective compression ratio of 7.5% (= 30% token retention $\times$ 25% bit-width reduction) while maintaining competitive performance (47.08 on LongBench, 90.28 on RULER-16K). This confirms that EntroKV and quantization compose effectively: the budget-level scheduling provided by EntroKV remains beneficial even when the underlying KV representations are aggressively quantized.

*Table 12.* Compatibility of EntroKV with INT4 KV-cache quantization (Llama-3.1-8B-Instruct).

| CONFIGURATION | LONGBENCH | RULER-4K | RULER-16K | RATIO (%) |
|---|---|---|---|---|
| ENTRO-SNAPKV (FP16) | 48.15 | 89.33 | 91.63 | 30.00 |
| 4-BIT QUANTIZATION ONLY | 47.20 | 90.72 | 90.40 | 25.00 |
| 4-BIT + ENTRO-SNAPKV | 47.08 | 88.04 | 90.28 | 7.50 |

## J.2. Entropy-Guided Mixed-Precision Quantization

Beyond composing with uniform quantization, the entropy signal can also guide *dynamic bit-width allocation* across heads—assigning higher precision to high-entropy (compression-sensitive) heads and lower precision to low-entropy heads.

**Experimental setup.**   Using a normalized attention entropy threshold of 0.5, we assign higher bit-widths to heads above the threshold ($\sim$15% of heads) and lower bit-widths to the rest ($\sim$85%). We compare entropy-guided mixed-precision allocation against random assignment matched to the same average bit-width and compression ratio.

*Table 13.* Entropy-guided mixed-precision quantization vs. uniform and random baselines (Llama-3.1-8B-Instruct, LongBench). Avg. bit-width is computed based on head proportions above/below the entropy threshold.

| SCHEME | LONGBENCH | AVG. BITS | RATIO (%) |
|---|---|---|---|
| FULL CACHE (FP16) | 48.26 | 16.00 | 100.00 |
| UNIFORM 8-BIT | 47.87 | 8.00 | 50.00 |
| UNIFORM 4-BIT | 47.20 | 4.00 | 25.00 |
| UNIFORM 2-BIT | 39.77 | 2.00 | 12.50 |
| 8/4-BIT (ENTROPY) | 47.75 | 4.60 | 28.75 |
| 8/4-BIT (RANDOM) | 47.18 | 4.60 | 28.75 |
| 4/2-BIT (ENTROPY) | 42.64 | 2.30 | 14.38 |
| 4/2-BIT (RANDOM) | 41.00 | 2.30 | 14.38 |

Under identical average bit-widths, entropy-guided allocation consistently outperforms random assignment (+0.57 for 8/4-bit, +1.64 for 4/2-bit), confirming that attention entropy is a reliable proxy for quantization sensitivity as well. This extends the core insight of our paper: entropy characterizes not only *token-level* compression difficulty (budget allocation) but also *precision-level* compression difficulty (bit-width allocation).

**Future directions.**   While these simulated experiments validate the algorithmic viability of entropy-guided mixed-precision, deploying true dynamic mixed-precision in practice requires customized GPU kernels for variable bit-width KV storage and retrieval. We consider the systematic study of entropy-guided mixed-precision quantization—including deployment-level kernel optimization and comprehensive baseline comparisons—as a promising direction for future work.

