# OpenReview forum: "EntroKV: Entropy-Guided Dynamic Budget Allocation for KV-Cache Compression"
_ICML.cc/2026/Conference — ICML 2026 spotlight_

### Official Review · Reviewer_jYTe · 2026-03-07

**Soundness:** 3
**Presentation:** 2
**Significance:** 3
**Originality:** 2
**Overall Recommendation:** 5
**Confidence:** 3

**Summary:**

This paper studies KV cache compression for long context LLM inference and presents that the main issue is not only how to compress, but also how to allocate a fixed retention budget across layers and heads. The proposed method, EntroKV, uses normalized attention entropy as an online signal to assign larger budgets to high entropy heads and smaller budgets to low entropy heads, with a calibrated global baseline and two control parameters. The method is presented as a front end allocator that can be attached to eviction and merging back ends such as SnapKV, CaM, and D2O. Experiments on LongBench, RULER, MATH 500, and SQuAD report better accuracy than fixed allocation baselines at similar memory budgets, with small overhead in memory and latency plots.

**Compliance With Llm Reviewing Policy:**

Affirmed.

**Final Justification:**

Thank the authors for the rebuttal. My concerns have been addressed. I have raised my score and encourage the authors to incorporate the new results and discussion into the revised version.

**Key Questions For Authors:**

Please check the three questions in the Weaknesses section.

**Limitations:**

Yes

**Strengths And Weaknesses:**

# Strengths
* The problem formulation is clear. This paper separates “how to compress” from “how much to compress,” and formalizes head-wise budget allocation under a global constraint.
* Useful efficiency analysis. For example, Figure 4 reports peak memory, prefill latency, decode latency, and batch decode latency.
* The plug and play nature makes the method practically valuable.

# Weaknesses
* The comparison experiments are a bit limited. Specifically, Llama 3.1 and Qwen2.5 are relatively old models. The authors are suggested to add results with more recent Llama and Qwen models.
* Explanations for surpassing Full Cache are not yet strong. Some EntroKV settings match or exceed Full Cache, and the paper explains this as possible denoising or suppression of noisy attention patterns. However, there is no direct evidence such as testing with targeted diagnostics on retained tokens, attention maps, or error cases.
* Missing discussion on related dynamic budget allocation works such as “LaCache: Ladder-Shaped KV Caching for Efficient Long-Context Modeling of Large Language Models (ICML 2025).”

---

> ### Author Rebuttal · Authors · 2026-03-28
>
> Dear Reviewer,
>
> We sincerely thank you for the constructive feedback and for recognizing the clarity, efficiency, and practical value of EntroKV. Below, we address your specific questions regarding model evaluations, empirical evidence, and more baselines.
>
> ---
>
> ## 1. Response to "Limited Models" and Evaluations on Recent Models
>
> To demonstrate generalizability, we evaluated EntroKV using **Qwen-3-8B** on LongBench, applying it as a frontend for various KV compression operators (SnapKV, CaM, D2O).
>
> | |LongBench|Ruler-4K|Ruler-16K|
> |-|-|-|-|
> |Full Cache|49.68|95.33|93.12|
> |SnapKV|48.25|93.47|91.21|
> |Entro-SnapKV|49.05|95.26|92.89|
> |CaM|48.84|93.58|91.24|
> |Entro-CaM|49.55|95.35|92.99|
> |D2O|48.51|93.31|91.41|
> |Entro-D2O|49.73|95.21|93.09|
>
> Results demonstrate that EntroKV generalizes to the recent model architectures, consistently enhancing fixed-allocation baselines.
>
> ---
>
> ## 2. Response to  "Observation on EntroKV exceeds Full Cache"
>
> We hypothesize long contexts contain substantial noise; moderate compression acts as an information bottleneck, filtering noise while preserving key evidence to sometimes outperform Full Cache.
>
> To provide empirical support, we analyzed the **2WikiMQA** dataset on Llama-3.1-8B-Instruct. We used Gemini 3 Pro to annotate the "gold evidence" spans and compared the attention maps and outputs of Full Cache versus EntroKV. (Due to the page and formatting constraints of the rebuttal, we cannot present the full set of visualization results here, full visualizations and case studies will be added to the Appendix). We identified two primary failure modes in Full Cache that EntroKV successfully resolves:
>
> **1. Misled by Distractors (Incorrect Extraction)**
>
> - **Full Cache Failure:** The model extracts incorrect but semantically related entities. Attention maps reveal that the overwhelming presence of redundant information makes it difficult for the model to clearly distinguish between the gold evidence and distractor paragraphs.
>
> - **EntroKV Solution:** By eliminating redundant information, EntroKV reduces background noise, enabling the model to better distinguish distractors from the gold evidence and precisely locate the correct answer.
>
>
> **2. Attention Dilution (Failure to Answer)**
>
> - **Full Cache Failure:** The model fails to find the answer. Attention maps reveal a "flat," highly diluted distribution across the massive sequence, drowning out the true evidence.
>
> - **EntroKV Solution:** Pruning irrelevant context forces the Softmax distribution to re-normalize over fewer tokens. This structurally amplifies the relative attention weights of the remaining useful information, allowing the model to confidently extract the answer.
>
>
> In summary, these attention dynamics confirm that EntroKV's dynamic allocation explicitly functions as an implicit denoiser, explaining its superior performance in certain long-context scenarios.
>
> ---
>
> ## 3. Response to "Supplementing Dynamic Budget Allocation Baselines":
>
> We thank the reviewer for pointing out "LaCache", which we will include in our "Related Work" section.
>
> However, we wish to respectfully clarify a technical nuance: **LaCache is fundamentally a pre-defined, structured KV cache pattern, rather than an input-dependent dynamic budget allocation method.** As stated in their paper, LaCache drops tokens based on a fixed "staircase" geometric pattern governed by static hyperparameters; it does not dynamically adjust budgets across different heads based on real-time contextual signals.
>
> **Expanded Baseline Comparisons:** Nevertheless, we fully agree with the core intent of your suggestion—comparing EntroKV with recent, genui snely "dynamic" budget allocation approaches is crucial. To this end, we conducted extensive new experiments on Llama-3.1-8B-Instruct, comparing EntroKV against several state-of-the-art dynamic allocation baselines, including **LAVa**, **Task-KV**, and **CAKE** [1,2,3].
>
> ||LongBench|Ruler-4K|Ruler-16K|
> |-|-|-|-|
> |Entro-SnapKV|**48.15**|**89.33**|**91.63**|
> |LaVA|47.25|85.24|87.12|
> |TaskKV|46.66|86.45|87.41|
> |CAKE|47.24|88.02|91.02|
>
> As demonstrated by the results, EntroKV achieves superior performance compared to these recent dynamic baselines. Furthermore, unlike LAVa (which requires computing a loss proxy) or Task-KV, EntroKV retains its unique zero-overhead, plug-and-play advantage by directly reusing forward-pass attention weights.
>
> > Our **rebuttal to reviewer r6kz** shows more comparison details.
>
> ---
>
> Thank you once again for your insightful suggestions, which have significantly strengthened the quality of our paper. Should you have any remaining questions, we look forward to further engaging with you during the discussion period.
>
> Authors
>
> ---
>
> ## Reference
> [1] LAVa: Layer-wise KV Cache Eviction with Dynamic Budget Allocation
>
> [2] Task-KV: Task-aware KV Cache Optimization via Semantic Differentiation of Attention Heads
>
> [3] CAKE: Cascading and Adaptive KV Cache Eviction with Layer Preferences

---

> > ### Author Rebuttal · Reviewer_jYTe · 2026-04-02
> >
> > Thank the authors for the rebuttal. My concerns have been addressed. I have raised my score and encourage the authors to incorporate the new results and discussion into the revised version.

---

> > > ### Author Response · Authors · 2026-04-03
> > >
> > > Dear Reviewer,
> > >
> > > Thank you for recognizing our work and raising your score. We will continue to improve our paper through revisions in the rebuttal phase. We greatly appreciate your time and effort in reviewing our manuscript.
> > >
> > > Best regards,
> > >
> > > Authors

---

### Official Review · Reviewer_LhZ1 · 2026-03-08

**Soundness:** 4
**Presentation:** 3
**Significance:** 3
**Originality:** 3
**Overall Recommendation:** 5
**Confidence:** 3

**Summary:**

The paper addresses the issue of static budget allocation in KV cache compression for Large Language Models (LLMs).
It introduces EntroKV, a plug-and-play framework that dynamically allocates memory budgets to different attention heads and layers based on their attention entropy.
By using entropy as a proxy for compression sensitivity, the method prioritizes budget for high-entropy (diffuse) heads while aggressively compressing low-entropy heads.
The authors validate their approach on long-context benchmarks (LongBench, Ruler), demonstrating performance improvements across various base compression operators (SnapKV, CaM, D2O).

**Compliance With Llm Reviewing Policy:**

Affirmed.

**Final Justification:**

The authors have satisfactorily addressed all of my concerns during the rebuttal period. Given the clarifications provided and the quality of the work, I vote Accept for this paper. This is a solid contribution that will be of interest to the community

**Key Questions For Authors:**

1. Clarification on Entropy Computation: Could you move the specifics of the entropy calculation (currently in Appendix E.1, Equation 9) into the main text of Section 3.3? Specifically, it would be helpful to explicitly state that it is computed over the observation window of the last w queries. Also, what if all queries from the prefill are used for the entropy calculation instead of an observation window?

2. Integration with Quantization: Do you have any preliminary experimental data combining EntroKV with a standard quantization method to empirically confirm the advantage of this method over other baseline budget allocations?

3. Calibration Sensitivity: How sensitive is the global baseline entropy ($\overline{H}$) to the specific data used in the calibration set? Would a user need to recompute this baseline if deploying the model in a highly specialized, non-general domain (e.g., purely medical or legal texts)?

**Limitations:**

yes

**Strengths And Weaknesses:**

**Strengths:**

`Strong Empirical Motivation`: The observation that attention recall varies drastically across different layer-head pairs and tasks is well-documented and provides a compelling justification for dynamic allocation.

`Plug-and-Play Design`: Decoupling the budget allocation strategy from the actual compression execution is a highly practical architectural choice.

`Experimental Results`: The experimental results are positive, consistent, and demonstrate that the approach effectively preserves model performance at high compression rates. I quite like the paper.

**Weaknesses & Feedback:**

`Clarity on Entropy Computation in Main Text`: While the paper proposes using the entropy of attention scores, the exact computation is somewhat glossed over in the main text. Appendix E.1 clarifies that it is computed over an observation window of the last $w$ queries and averaged (Equation 9). Given that this is the core metric of the paper, the authors should explicitly define which queries are used and how the averaging is handled in Section 3.3 to avoid ambiguity for the reader.

`Lack of Quantization Experiments`: The experiments successfully demonstrate compatibility with eviction and merging-based operators. However, quantization is entirely missing from the evaluation. Since quantization is a massively popular KV-cache compression technique, it would significantly strengthen the paper to include at least one experiment combining EntroKV with a quantization method to empirically prove this orthogonality and ensure there are no unforeseen compounding errors.

---

> ### Author Rebuttal · Authors · 2026-03-28
>
> Dear Reviewer,
>
> We sincerely thank you for the positive assessment and for highlighting the strong empirical motivation and practical plug-and-play design of EntroKV. We deeply appreciate your constructive feedback, which will help strengthen the clarity and empirical rigor of our paper. Below, we address your specific questions.
>
> ---
>
> ## 1. Clarification on Entropy Calculation and the Use of the Observation Window
>
> As suggested, we will move the exact entropy calculation formula (currently Eq. 9, Appendix E.1) into **Section 3.3** to eliminate ambiguity. And we think we need to clarify that using only the last $w$ queries (rather than all prefill queries) aligns with SnapKV's mechanism and maintains strict Flash-Attention compatibility.
>
> - **Hardware and Memory Constraints:** Calculating entropy for all queries in the prefill stage would force the system to materialize the full $N \times N$ attention matrix. This would lead to an unacceptable surge in GPU memory usage, defeating the original purpose of efficient long-context inference. By restricting the calculation to an observation window, we successfully bypass the $O(N^2)$ memory bottleneck (Let $N$ be the total input length and $w$ the observation window, and $N \gg w$ ).
>
> - **Ablation Studies on Window Length:** During our exploration, we conducted ablation studies on the length of this observation window. We found that using a smaller number of tokens for $w$ already provides a highly reliable proxy for capturing the inherent attention dispersion (i.e., compression sensitivity) of a specific head, as well as serving as an indicator for the importance of preceding tokens. We will include a brief discussion of these ablation results in the revised version.
>
> w|128|64|32|16|8
> -|-|-|-|-|-
> socre|47.98|47.92|48.03|47.98|47.87
>
> - **Generalization Capability:** We would also like to emphasize that the observation window based on $w$ queries is merely a specific instantiation adapted for SnapKV. For other compression operators, the entropy calculation can be fully transformed according to their respective attention mechanisms. This allows for compatibility with minimal additional overhead, thereby preserving the **plug-and-play** nature of our framework. We will supplement the main text with these critical system-level details.
>
> ---
>
> ## 2. Integration with Quantization
>
> As discussed in the paper, EntroKV serves as a budget allocation frontend for eviction and merging operators. Theoretically, it remains fully orthogonal to representation-layer quantization methods. To empirically validate this orthogonality, we conducted additional compatibility experiments by integrating EntroKV with standard **INT4 KV-cache quantization**. These configurations were evaluated on the LongBench and Ruler benchmarks.
>
> ||LongBench|Ruler 4K|Ruler 16K|Compression Ratio(%)|
> |-|-|-|-|-|
> |entrokv|48.15|89.33|91.63|30.00|
> |4bit_quan|47.20|90.72|90.40|25.00|
> |4bit_quan + entrokv|47.08|88.04|90.28|7.50|
>
> The results confirm that EntroKV maintains robust performance even under high compression ratios when paired with 4-bit quantization, effectively demonstrating the **orthogonality** between the two approaches.
>
> ---
>
> ## 3. Sensitivity to Global Baseline ($\overline{H}$) Calibration
>
> We would like to clarify that the global baseline $\overline{H}$ is specifically designed to aggregate diverse data distributions, thereby providing a stable **reference anchor**.
>
> As briefly illustrated in **Figure 6(c)**, attention entropy inherently reflects task-specific information density: low-density tasks (e.g., Document QA, Information Retrieval) exhibit lower entropy, while high-density tasks (e.g., Summarization, Code) show higher entropy. The calibration set intentionally blends these diverse data types to derive a median anchor. This anchor is crucial for preventing **dynamic allocation drift**—relying solely on real-time signals can lead to distribution imbalances and subsequent performance degradation, as evidenced by our ablation study on the adaptive coefficient $\alpha$ in **Figure 5(a)**. Consequently, the baseline is architecturally robust and highly generalizable.
>
> Naturally, for deployment in extreme, highly specialized vertical domains (e.g., pure medical or legal corpora where global information density shifts fundamentally), recalibrating this anchor can be beneficial. Fortunately, as detailed in **Appendix D**, this process is **computationally lightweight**, requiring only approximately 500 samples. We will include a dedicated discussion on this task-specific calibration mechanism in the revised manuscript.
>
> ---
>
> We hope these clarifications and the additional experimental results fully address your questions. As suggested, we will incorporate the detailed entropy computation into Section 3.3, add the quantization results, and expand our discussion on calibration in the final version of the manuscript. Thank you again for your valuable time and insights.
>
> Authors

---

> > ### Author Rebuttal · Reviewer_LhZ1 · 2026-04-01
> >
> > I thank the authors for their responses. To clarify a point of discussion: my question regarding the combination of EntroKV and quantization appears to have been slightly misunderstood. Specifically, I was suggesting a dynamic bit-width allocation strategy where the quantization budget for each KV cache is determined by EntroKV’s importance metrics, rather than applying a uniform 4-bit quantization. I would appreciate the authors' thoughts on the feasibility of such an integrated approach.

---

> > > ### Author Response · Authors · 2026-04-01
> > >
> > > Dear Reviewer,
> > >
> > > Thank you for your further clarification. We greatly appreciate the insightful proposal: the attention entropy metric of EntroKV should not only guide token retention budgets but also serve as a signal for dynamic mixed-precision bit-width allocation (e.g., assigning 4-bit/8-bit to high-entropy heads and 2-bit to low-entropy heads).
> > >
> > > This perfectly aligns with the core philosophy of our paper. In response, we first designed a mini-experiment to validate this quantization strategy. Subsequently, we conducted an in-depth analysis of its algorithmic feasibility and system deployment complexity.
> > >
> > > ---
> > >
> > > ### 1. Mini-Experiment
> > >
> > > - **Basic Idea**: Since deploying true dynamic mixed-precision requires complex, customized GPU kernels, we adopt a **quantize-then-dequantize** paradigm (simulated in FP16) to emulate the algorithmic accuracy loss. Using a predefined Normalized Attention Entropy threshold of 0.5, we assign higher bit-widths to heads above the threshold and lower bit-widths to the rest.
> > >
> > > - **Experimental Setup**: Using Llama-3.1-8B-Instruct, we evaluate the performance of Entropy-guided mixed-precision (8/4-bit and 4/2-bit) against uniform quantization baselines. **Statistically, the proportion of heads above the 0.5 threshold is approximately 0.15, while those below account for 0.85. The Avg. Bit-Width in the table is calculated based on these proportions (e.g., $8 \times 0.15 + 4 \times 0.85 = 4.60$).** For a fair comparison, we introduce a **random** assignment control group that strictly matches the average equivalent bit-width and total compression ratio of our Entropy-guided scheme.
> > >
> > > - **Experimental Results**:
> > >
> > > |**Quantization Scheme**|**LongBench Score**|**Avg. Bit-Width**|**Compression Ratio (%)**|
> > > |---|---|---|---|
> > > |w/o quan (Full Cache)|48.26|16.00|100.00|
> > > |Only 8bit|47.87|8.00|50.00|
> > > |Only 4bit|47.20|4.00|25.00|
> > > |Only 2bit|39.77|2.00|12.50|
> > > |**8bit & 4bit (entropy)**|47.75|4.60|28.75|
> > > |8bit & 4bit (random)|47.18|4.60|28.75|
> > > |**4bit & 2bit (entropy)**|42.64|2.30|14.38|
> > > |4bit & 2bit (random)|41.00|2.30|14.38|
> > >
> > > ---
> > >
> > > ### 2.  Discussion on Algorithmic Feasibility
> > >
> > > The experimental results clearly validate the algorithmic viability of your proposal:
> > >
> > > - **Entropy is a reliable proxy for quantization sensitivity:** Under identical average bit-widths and compression ratios, the entropy-guided allocation outperforms random assignment. This confirms that entropy effectively preserves model accuracy during dynamic bit-width allocation.
> > >
> > > - **Extension of our theoretical framework:** Our main paper establishes entropy as a signal for token-level compression (characterizing information density), originally treating precision quantization as an orthogonal method. These new results extend this insight: entropy serves as a general proxy for overall compression difficulty, demonstrating strong performance in guiding information-precision quantization.
> > >
> > > ---
> > >
> > > ### 3. Future Work: Deployment Challenges and Systematic Comparisons
> > >
> > > Although our small-scale experiment validates that dynamic allocation effectively preserves model accuracy, the mixed-precision scheme poses substantial deployment challenges for real-world AI infrastructure in more complex practical scenarios. Additionally, a systematic comparison with other mature KV Cache quantization methods is also lacking. This issue naturally goes beyond the scope of the current paper, and we consider it a promising direction for future work.
> > >
> > > ---
> > >
> > > In our revised manuscript, we will include the details of this simulated experiment in the Appendix. Additionally, we will explicitly highlight entropy-guided dynamic mixed-precision quantization—along with its deployment challenges and systematic baseline comparisons—as a key direction for future research. We sincerely thank you for this forward-looking feedback, which has greatly broadened the scope of our work.
> > >
> > > Sincerely,
> > >
> > > Authors

---

### Official Review · Reviewer_r6kz · 2026-03-08

**Soundness:** 3
**Presentation:** 3
**Significance:** 3
**Originality:** 2
**Overall Recommendation:** 4
**Confidence:** 3

**Summary:**

In EntroKV, the authors proposed a new KV-cache compression method for long-context LLM inference. The key idea is that we can use entropy as an online proxy to determine the budget allocation across layers and heads. High-entropy heads generally are more sensitive to compression and down-stream accuracy, thus should be allocated with more budgets. The experiments show that EntroKV outperforms the previous unified-budget solutions, achieving comparable results to full-cache performance while significantly saving GPU memories.

**Compliance With Llm Reviewing Policy:**

Affirmed.

**Final Justification:**

Rebuttal addressed my main concerns.

**Key Questions For Authors:**

Please check the weakness part. Can authors include the comparison and discussion with the similar dynamic budget allocation method?

Figure2 (c) shows high-entropy heads correlate with high importance. How are the importance calculated? What are the tasks? Does this relationship generalize across different tasks?

In the evaluation, the author mentioned that EntroKV can exceeds full-cache accuracy by filtering noise, which is contradictory to the analysis in Section 3.1 that allocating more KV capacity is non-increasing in terms of performance. Can you better substantiate this claim?

**Limitations:**

yes

**Strengths And Weaknesses:**

In generally, compressing KV-cache is of practical significance and allocating the budgets dynamically can result in better performance while saving the memory. The motivations, the explanation of the designs and evaluation results are all sound to me.

My major concern is the lack of the compression to similar dynamic budget methods. For example, LAVa [https://arxiv.org/abs/2509.09754] uses layer attention output loss for dynamic head budget allocation. If entropy, which is the metric used by the authors, is a more strong online proxy, can it achieve better performance in the same setting? Similarly, MEDA [https://arxiv.org/pdf/2502.17599] use cross-modal entropy, Task-KV [https://arxiv.org/html/2501.15113v1] use semantic difference for for budget allocation. I believe the discussion and comparison between these similar methods are essential to verify the claims made by the authors.

---

> ### Author Rebuttal · Authors · 2026-03-28
>
> Dear Reviewer,
>
> We sincerely thank you for recognizing the significance of our work and for your constructive feedback. Below, we address your specific questions and detail the new experiments incorporated.
>
> ## 1. Response to "Additional Dynamic Allocation Baselines"
>
> We evaluated additional methods on Llama-3.1-8B-Instruct. We exclude **MEDA** because it is designed for multimodal settings, while our study focuses on NLP only. We instead include **CAKE**[1] as a new baseline.
>
> **Direct downstream comparison** (to be added to **Table 2** with PyramidKV and AdaKV):
>
> |Method|LongBench|Ruler-4K|Ruler-16K|
> |-|-|-|-|
> |Entro-SnapKV|**48.15**|**89.33**|**91.63**|
> |LAVa|47.25|85.24|87.12|
> |TaskKV|46.66|86.45|87.41|
> |CAKE|47.24|88.02|91.02|
>
> **Why entropy is a stronger proxy?** We compare methods by proxy signal, allocation granularity (head/layer/task), and extra allocation operations, highlighting where EntroKV differs from prior methods in both allocation scope and implementation complexity.
>
> |Method|Proxy Signal|Head-level|Layer-level|Task-level|Extra Allocation Operations|
> |-|-|-|-|-|-|
> |**PyramidKV**|Layer-Depth Prior|✗|✓|✗|Fixed depth-based layer schedule lookup|
> |**AdaKV**|Accumulated Attention Mass|✓|✗|✗|Per-head attention-mass accumulation and selection|
> |**LAVa**|Output-Loss Surrogate *(recent attention × value norm)*|✓|✓|✗|Recent-attention/value-norm statistics and ranking|
> |**Task-KV**|Semantic Differentiation *(distance to semantic center)*|✓|✗|✓|Semantic-center distance computation and head partitioning|
> |**CAKE**|Layer Preference *(entropy + temporal variance)*|✗|✓|✗|Layer entropy/variance statistics and cascading redistribution|
> |**EntroKV**|Normalized Attention Entropy|✓|✓|✓|Single entropy reduction over existing attention weights|
>
> And we summarize the strengths of EntroKV as follows:
>
> - **Length-normalized and directly comparable**: EntroKV uses normalized attention entropy (**eq. (4)**), which makes scores directly comparable across heads, layers, and inputs.
> - **No auxiliary branch at inference**: EntroKV reuses attention weights already materialized in the SnapKV observation window and only adds a lightweight reduction, with no additional matrix multiplications.
> - **Theoretically motivated**: Under the assumptions in **Section 3.2**, entropy is Schur-concave and monotonically aligned with Top-K recall difficulty (**Appendix F**), giving a principled link between attention dispersion and allocation demand.
>
> ## 2. Response to "Calculation of Importance Scores in Figure 2(c)"
>
> The importance score is Taylor Importance (**Eq. 3**), which quantifies a component's contribution by measuring the change in model loss when masked. To ensure generalizability, we combine diverse tasks from LongBench and RULER, select 5 samples per task, and sample 1024 points in total.
>
> To analyze task-specific differences, we compute the **Pearson correlation coefficient** between high entropy and high importance:
>
> |Dataset|hotpotqa|niah_1|lcc|cwe|
> |-|-|-|-|-|
> |R|0.47|0.39|0.56|0.55|
>
> Coding(lcc) and common word extraction(cwe) show stronger correlation, while Needle-In-A-Haystack(niah_1) and document QA(hotpotqa) are weaker. We hypothesize that for tasks requiring global context, high-entropy attention heads are even more essential for effective information aggregation.
>
>
> ## 3. Response to "the Conflict Between Performance Exceeding Full Cache and Theoretical Modeling"
>
> We sincerely thank the reviewer for this highly meticulous observation. We would like to respectfully clarify that there is no mathematical contradiction. This misunderstanding stems from a subtle but crucial distinction between microscopic approximation error (Section 3.1) and macroscopic downstream task performance.
>
> - **Microscopic vs. Macroscopic:** The monotonic non-increasing assumption for local compression loss $\mathcal{L}_l^h(B)$ correctly dictates that a larger budget helps a single head approximate its Full Cache behavior. Yet, perfectly reconstructing Full Cache for all heads does not guarantee optimal downstream accuracy, as LLMs are highly susceptible to noise in long contexts (e.g., Lost in the Middle [2]).
>
> - **Implicit Denoising:** EntroKV allocates minimal budget to low-entropy (often noisy/redundant) heads, which prevents reconstructing noisy Full Cache behavior and implicitly improves macroscopic downstream performance.
>
> - **Empirical Evidence:** Figure 2(a) shows that at the same attention recall, a larger global budget rather produces higher downstream loss, indicating that forcing full attention recovery may introduce harmful noise.
>
> > For additional evidence on cases where EntroKV exceeds Full Cache, please refer to our **rebuttal to Reviewer jYTe**.
>
> We hope these additions clarify the baselines, importance-score computation, and the theoretical interpretation.
>
> Authors
>
> ### References
>
> [1] CAKE: Cascading and Adaptive KV Cache Eviction with Layer Preferences
>
> [2] Lost in the Middle: How Language Models Use Long Contexts

---

> > ### Author Rebuttal · Reviewer_r6kz · 2026-04-02
> >
> > Thanks for the feedback. I will adjust my review score.

---

> > > ### Author Response · Authors · 2026-04-03
> > >
> > > Dear Reviewer,
> > >
> > > Thanks for your positive acknowledgment. Best wishes!
> > >
> > > Sincerely,
> > >
> > > Authors

---

### Official Review · Reviewer_GZri · 2026-03-14

**Soundness:** 3
**Presentation:** 3
**Significance:** 3
**Originality:** 3
**Overall Recommendation:** 4
**Confidence:** 4

**Summary:**

This paper proposes EntroKV, a dynamic budget allocation framework for KV-cache compression in large language model inference. The key idea is to allocate KV-cache retention budgets adaptively across layers and attention heads based on attention entropy, which is used as a proxy for compression sensitivity. The authors argue that high-entropy heads require larger retention budgets while low-entropy heads can be aggressively compressed. EntroKV functions as a lightweight scheduling module that can be combined with different compression operators (e.g., eviction or merging). Experiments across several tasks show that the proposed approach consistently improves performance under constrained KV-cache budgets, retaining around 98% of full-cache performance at a 30% budget ratio with minimal overhead.

**Compliance With Llm Reviewing Policy:**

Affirmed.

**Final Justification:**

The new evidence addresses my previous concerns. Given the improved empirical performance, I have increased my score from 3 to 4. I encourage the authors to incorporate these new results and discussions into the final version of the paper.

**Key Questions For Authors:**

- The paper only shows the results of 16k context length. How about longer length such as 32k? Whether EntroKV maintains strong performance under longer contexts remains unverified.
- It would be valuable to evaluate whether the method also performs well on newer models such as Qwen3-8B.

**Limitations:**

yes

**Strengths And Weaknesses:**

**Strengths**

- The method is designed as a plug-and-play module, making it compatible with different existing compression operators. This modularity improves practical usability.
- Experiments show consistent improvements over static allocation baselines under constrained KV budgets.

**Weakness**

- The core idea of using entropy as an importance proxy is relatively simple, and the novelty compared to existing dynamic KV-cache allocation methods appears somewhat incremental.
- The paper focuses primarily on moderate context lengths and does not evaluate performance under substantially longer contexts (e.g., 32k or beyond). Since KV-cache compression becomes particularly important in such regimes, it would be valuable to demonstrate whether the proposed allocation strategy remains effective when context length scales further.
- The missing of **Impact Statement**, which is required on [Call for Papers](https://icml.cc/Conferences/2026/CallForPapers)

---

> ### Author Rebuttal · Authors · 2026-03-28
>
> Dear Reviewer,
>
> We sincerely appreciate your recognition of our plug-and-play design and consistent performance gains. To directly address your feedback regarding our method's simplicity, longer context scaling, and newer models, we provide theoretical clarifications and extensive new experiments below.
>
> ## 1. Response to "Concerns on Novelty and Simplicity"
>
> Regarding the concerns on design simplicity and perceived incremental innovation, we would like to offer the following clarifications:
>
> ### 1.1. From an Entropy Signal to a Constrained Resource Allocation Framework
>
> Our contribution extends beyond merely adopting "entropy." We formulate KV cache compression as a **Constrained Resource Allocation** problem (**Section 3.1**). As detailed in **Appendix F**, minimizing compression loss mathematically requires equalizing the **Marginal Recall Utility** across attention heads. Entropy, while initially informed by empirical observation, is rigorously proven to be a theoretically self-consistent proxy within our mathematical framework. Thus, the simplicity of the final algorithm is a natural consequence of the formal derivation, distinguishes our approach from purely heuristic designs.
>
> ### 1.2. Limitations of Current Allocation Strategies and EntroKV’s Mitigation
>
> Existing allocation baselines (e.g., PyramidKV, AdaKV [1,2]) often rely on static structural heuristics—such as assuming lower layers invariably require larger budgets. Faced with the complexity of semantics and the opacity of LLM internals, this inflexible assumption causes severe resource mismatch. As shown in **Figure 6(b)**, the information density of the KV cache does **not** decrease monotonically with increasing layer depth, and their relationship is difficult to characterize using heuristic functions. Therefore, it is necessary to introduce a real-time proxy metric.
>
> By using real-time entropy as a proxy, EntroKV dynamically allocates budgets based on actual information flow, fundamentally avoiding the limitations of static layer-bound assumptions.
> > We further provide a more detailed, method-level comparison in our **rebuttal to Reviewer r6kz**.
>
> ### 1.3. Why Simplicity Matters: System-Level Efficiency and Deployability
>
> We would also like to highlight that in high-concurrency LLM inference, algorithmic minimalism is a system-level requirement rather than a compromise. Our design reuses existing attention weights, introducing zero additional matrix multiplications. To demonstrate the practical applicability of this design, we evaluated performance in batched decoding scenarios in **Figure 4**. We note that some dynamic methods (e.g., LAVA, AdaKV [2,3]) do not include evaluations in such settings. We believe that more complex dynamic mechanisms may face additional challenges when adapting to high-concurrency inference environments, though we acknowledge that this deserves further systematic study.
>
> ---
>
> ## 2. Response to "Evaluation on Longer Contexts (e.g., 32k or longer)"
>
> We have extended our evaluation to a **128K context length** on the RULER benchmark (Llama-3.1-8B-Instruct).
>
> ||32K|64K|128K|
> |-|-|-|-|
> |Full Cache|88.16|84.90|75.45|
> |SnapKV|86.83|82.40|72.68|
> |Entro-SnapKV|87.19|84.24|74.00|
>
> As shown, EntroKV consistently outperforms the SnapKV baseline as context length increases, proving that entropy remains a highly resilient scheduling signal even under extreme context scaling (128K).
>
> ---
>
> ## 3. Response to "Evaluation on Newer Models (e.g., Qwen-3-8B)"
>
> Following your suggestion, we evaluated EntroKV on **Qwen-3-8B**(LongBench and RULER benchmarks), acting as a front-end module for both eviction (SnapKV) and merging (CaM, D2O) operators.
>
> | |LongBench|Ruler-4K|Ruler-16K|
> |-|-|-|-|
> |Full Cache|49.68|95.33|93.12|
> |SnapKV|48.25|93.47|91.21|
> |Entro-SnapKV|49.05|95.26|92.89|
> |CaM|48.84|93.58|91.24|
> |Entro-CaM|49.55|95.35|92.99|
> |D2O|48.51|93.31|91.41|
> |Entro-D2O|49.73|95.21|93.09|
>
> The results demonstrate that EntroKV seamlessly generalizes to the newer architectures, consistently enhancing underlying compression operators. These results will be incorporated into our final manuscript.
>
> ---
>
> ## 4. Response to "Missing Impact Statement"
>
> We apologize for this oversight. In the camera-ready version, we will include the standard "Impact Statement" recommended by the conference guidelines, noting that our research focuses on foundational LLM inference efficiency without introducing specific ethical risks.
>
> ---
>
> We hope these additional results and clarifications effectively address your concerns. Should you have any remaining questions, we look forward to further engaging with you during the discussion period.
>
> Authors
>
> ---
>
> ## Reference
> [1] PyramidKV: Dynamic KV Cache Compression based on Pyramidal Information Funneling
>
> [2] Ada-KV: Optimizing KV Cache Eviction by Adaptive Budget Allocation for Efficient LLM Inference
>
> [3] LAVa: Layer-wise KV Cache Eviction with Dynamic Budget Allocation

---

> > ### Author Rebuttal · Reviewer_GZri · 2026-04-02
> >
> > Thank you for the detailed rebuttal and for providing the additional experimental results. The new evidence addresses my previous concerns. Given the improved empirical performance, I have increased my score from 3 to 4. I encourage the authors to incorporate these new results and discussions into the final version of the paper.

---

> > > ### Author Response · Authors · 2026-04-03
> > >
> > > Dear Reviewer,
> > >
> > > We sincerely appreciate your positive acknowledgment of our work and the increased score. We believe the modification  during the rebuttal stage will further strengthen the quality and clarity of our paper. Thank you again for your valuable time and constructive feedback on our manuscript.
> > >
> > > Best regards,
> > >
> > > Authors

---

### Decision · Program_Chairs · 2026-04-30

**Decision:**

Accept (spotlight)

**Comment:**

This paper proposes a dynamic budget allocation framework based on attention entropy for KV-cache compression in large language model inference. The paper received consistently high evaluations from 4 reviewers. There have been abundant discussions during the rebuttal, and the authors' responses have addressed all the concerns.